# Discovery and characterization of potent pan-variant SARS-CoV-2 neutralizing antibodies from individuals with Omicron breakthrough infection

Yu Guo [1,2,3,4,10] ✉, Guangshun Zhang[1,3,5,6,10], Qi Yang [3,10] ✉, Xiaowei Xie[2,10], Yang Lu[2,10], Xuelian Cheng[2,10], Hui Wang[4,5,10], Jingxi Liang[1,7,10], Jielin Tang [3,10], Yuxin Gao[1,5,6,10], Hang Shang[1,5,6], Jun Dai[8], Yongxia Shi[8], Jiaxi Zhou [2], Jun Zhou [1,2], Hangtian Guo[7], Haitao Yang [7], Jianwei Qi[2], Lijun Liu[2], Shihui Ma[2], Biao Zhang[2], Qianyu Huo[2], Yi Xie [9], Junping Wu [9], Fang Dong[1,6], Song Zhang[1,6], Zhiyong Lou [3], Yan Gao [7], Zidan Song[1,3,5,6], Wenming Wang[1,5,6], Zixian Sun[3], Xiaoming Yang [4,5] ✉, Dongsheng Xiong[2] ✉, Fengjiang Liu [3] ✉, Xinwen Chen[3] ✉, Ping Zhu [2] ✉, Ximo Wang[9] ✉, Tao Cheng [2] ✉ & Zihe Rao [1,3,5,7] ✉

The SARS-CoV-2 Omicron variant evades most currently approved neutralizing antibodies (nAbs) and caused drastic decrease of plasma neutralizing activity elicited by vaccination or prior infection, urging the need for the development of pan-variant antivirals. Breakthrough infection induces a hybrid immunological response with potentially broad, potent and durable protection against variants, therefore, convalescent plasma from breakthrough infection may provide a broadened repertoire for identifying elite nAbs. We performed single-cell RNA sequencing (scRNA-seq) and BCR sequencing (scBCR-seq) of B cells from BA.1 breakthrough-infected patients who received 2 or 3 previous doses of inactivated vaccine. Elite nAbs, mainly derived from the IGHV2−5 and IGHV3-66/53 germlines, showed potent neutralizing activity across Wuhan-Hu-1, Delta, Omicron sublineages BA.1 and BA.2 at picomolar $NT_{50}$ values. Cryo-EM analysis revealed diverse modes of spike recognition and guides the design of cocktail therapy. A single injection of paired antibodies cocktail provided potent protection in the K18-hACE2 transgenic female mouse model of SARS-CoV-2 infection.

The emergence of SARS-CoV-2 has caused >755 million confirmed cases and over 6.8 million deaths globally in the past three years[1]. Moreover, SARS-CoV-2 continues to evolve during the pandemic, giving rise to an increasing number of variants, including the alpha, beta, gamma, and delta variants of concern (VOCs)[2]. Omicron, an emerging variant reported in South Africa in November 2021, was soon designated the fifth VOC[3] by the World Health Organization (WHO) and replaced the Delta variant as the globally dominant variant in less

A full list of affiliations appears at the end of the paper. ✉e-mail: guoyu@nankai.edu.cn; yang_qi@gzlab.ac.cn; yangxiaoming@sinopharm.com; dsxiong@ihcams.ac.cn; liu_fengjiang@gzlab.ac.cn; chen_xinwen@gzlab.ac.cn; zhuping@ihcams.ac.cn; wangximo@126.com; chengtao@ihcams.ac.cn; raozh@mail.tsinghua.edu.cn

than 3 months, posing a severe challenge to the effectiveness of pro-phylactic and therapeutic interventions[4,5].

The spike (S) protein is the key component of the vaccine, and compared with the ancestral strain, the Omicron strain contains 34 distinct insertions, deletions, and mutations[6], along with 15 amino acid mutations located in the receptor-binding domain (RBD), which leads to widespread escape from drugs and vaccines developed based on the ancestral strain in the early stage[7,8], especially RBD-specific antibodies[6]. Notably, the Omicron variant has also evolved into multiple sublineages. BA.2.12.1, BA.4, and BA.5 are closely related to BA.2[5], with D405N and R408S mutations and additional L452Q, L452R, and F486V mutations[4,5]. BA.1 infection induces BA.1-specific neutralizing antibodies, which are also mostly escaped by BA.2.12.1, BA.4, and BA.5[4,9]. Although the most commonly administered mRNA vaccine retains some neutralizing activity, it also exhibits significant immune evasion against emerging VOCs such as BA.5, BQ.1.1, and XBB.1[9,10].

Recently, there have been serious concerns about the long-term sequelae of COVID-19, where a substantial percentage of patients continue to experience symptoms of a physical, psychological, or cognitive nature that are seen as the next public health disaster[11]. As previously reported, nAb therapy has been shown to be an important countermeasure to prevent and treat SARS-CoV-2, especially for elderly individuals who have difficulty developing neutralizing antibodies after vaccination or individuals with preexisting conditions who cannot be vaccinated[12]. However, the majority of the currently approved SARS-CoV-2 nAbs and cocktail therapy have been reported to reduce or abrogate neutralizing potency against emerging VOCs, especially for BQ.1.1 and XBB[13,14]. such as LY-CoV1404[15], REGN10933 + REGN10987[16], LY-CoV016+LY-CoV555[17,18], and AZD1061 + AZD8895[19]. Therefore, there is an urgent need to develop a new generation of broadly neutralizing antibodies to protect vulnerable populations from new emerging VOCs.

The discovery and molecular understanding of broad-spectrum, potent nAbs have advanced the development of nAb therapy and the design of next-generation recombinant vaccines. Recent studies have reported that Omicron breakthrough infection or a vaccine booster elicits an efficient humoral response, but due to immune imprinting, it is likely to be directed primarily at the epitopes of ancestral strains[8,20–22], nevertheless, studies have also reported that breakthrough infection can elicit more potent, breadth, and durability of serum-neutralizing responses[23]. Hence, the humoral response of inactivated vaccinees after breakthrough infection deserves further study. On the other hand, we believe that breakthrough infection-derived convalescent serum could also serve as a broadened repertoire for the discovery of elite antibodies. In this study, we used single-cell sequencing technology to isolate hundreds of monoclonal antibodies from convalescent patients of BA.1 breakthrough infection and identified dozens of elite antibodies with $NT_{50}$ values at the picomolar level. We performed cryo-EM studies on select antibodies to reveal the structural basis of broad and potent neutralization. Based on the structural mechanisms, we rationally designed antibody cocktails that showed excellent neutralizing activity against all SARS-CoV-2 VOCs including BQ.1.1 and XBB, providing a promising candidate for future development.

## Results

### Identification of efficient clonotypes by antigen-enriched single-cell RNA-seq coupled with single-cell BCR-seq

Considering the rapid spread and severe immune evasion of Omicron, we attempted to enrich broad-spectrum neutralizing antibodies against SARS-CoV-2, especially the Omicron variant. For this purpose, a single-cell transcriptomic profile of B cells derived from 38 infectious Omicron patients was obtained by 10× Genomics. Magnetic beads with a biotinylated BA.1 RBD were applied to purify antigen-binding B cells. Simultaneously, single-cell VDJ sequencing was also performed for each RBD+ B-cell (Fig. 1a). To accurately identify memory B cells

(MBCs), a total of 277,630 B cells were analyzed using the Atlas of Blood Cells (ABC), a comprehensive transcriptomic landscape involving 32 human blood cell types. Both annotated cell types and the UMAP visualization of canonical signature genes contributed to confirming the identification of MBC. We observed that CD27 and GPR183 were highly expressed in MBCs, while naive B cells were characterized by high expression of TCL1A, IL4R, and IGHD (Fig. 1b).

Next, clonotypes were constituted by dominant heavy and light chains according to previous reports[24,25]. In total, 204,124 clonotypes with distinct VDJ sequences were detected for further selection (Fig. 1c). Then, a series of filter criteria were adopted to elevate the precision of detecting neutralizing antibodies. The clonotypes featuring a frequency higher than 1, expressing IGHG1 but not IGHG2, having a somatic hypermutation (SHM) rate higher than 2% (corresponding to sufficient affinity maturation), and containing at least 1 MBC were considered to have high confidence in subsequent experiments[24] (Fig. 1c). Among these 286 high-confidence candidates, we observed that clonal expansion, number of MBCs and SHMs of most clonotypes were relatively inferior (Fig. 1d). Analysis of VDJ genes showed that IGHV2−5/IGHV3−23/IGHV1−69D for the heavy chain and IGLV2−14/IGLV1−40 for the light chain frequently participated in the VDJ gene rearrangements in Omicron patients (Fig. 1e, f, S1a, b). Additionally, the frequency and number of memory B cells in vaccinated patients were significantly increased compared with those in unvaccinated patients (Fig. S1c, d), while their SHM rates were significantly reduced (Fig. S1e), implying efficient immunological memory after prior vaccination. It is worth noting that by using BA.1 RBD as bait, we observed the preferential use of IGHV2−5 in vaccinated patients, whereas IGHV3−33/3−7 frequently participated in VDJ gene rearrangements in unvaccinated patients (Fig. S1f), which was consistent with a previous report that the IGHV2−5 lineage represented one of the best germlines for broadly neutralizing antibodies[15,26,27]. However, the antibodies derived from IGHV3−66/3−53 or IGHV3−33 reported earlier[12,28] that target the receptor-binding motif (RBM) region was therefore easily escaped by Omicron variants. In summary, 286 high-confidence clonotype candidates with a usage preference for IGHV2−5 were identified in infectious Omicron patients by single-cell transcriptomic and immune profiling.

### Discovery and preliminary characterization of neutralizing antibodies

Next, all 286 antibodies were expressed in HEK293 cells by transfection. To narrow the range of candidates, we evaluated the cross-binding and broadly blocking abilities of the mAbs against various types of SARS-CoV-2 by ELISA. Out of 286 antibodies, a total of 119 candidates achieved saturated absorption (OD450 > 4) against WT S protein, Omicron S protein, and Omicron RBD at the same time (Fig. 2a, S2a). Among those broadly binding mAbs, 44 mAbs could effectively block ACE2 interaction with multiple VOCs (Fig. 2a, S2b).

To examine the neutralizing activity of these 44 antibodies, we performed neutralization assays on authentic SARS-CoV-2 variants. The neutralization potency of these monoclonal antibodies spanned the single-digit to sextuple-digit ng/mL range against Wuhan-Hu-1, Delta, BA.1.1 and Omicron BA.2 variants (Fig. 2a–c, S3a, b). A previously reported elite nAb, XGv347, was used as a positive control (PC)[26]. Nineteen out of the 44 monoclonal antibodies with geometric mean half-maximal neutralizing titers ($NT_{50}$) less than 150 ng/mL against Omicron BA.2 variants were selected (Fig. 2a, S3a, b), and the features of these 19 optimized clonotypes were analyzed and summarized (Fig. S4a–c). Fourteen out of 19 monoclonal antibodies showed broad neutralizing ability against the Wuhan-Hu-1 strain and VOCs (Fig. 2b, c). Compared with the Wuhan-Hu-1 and Delta strains, the neutralization potencies of these 19 monoclonal antibodies against Omicron BA.1.1 and BA.2 were 10.7-fold and 4-fold lower on average, respectively (Fig. 2c). Among the 19 monoclonal antibodies, the neutralization

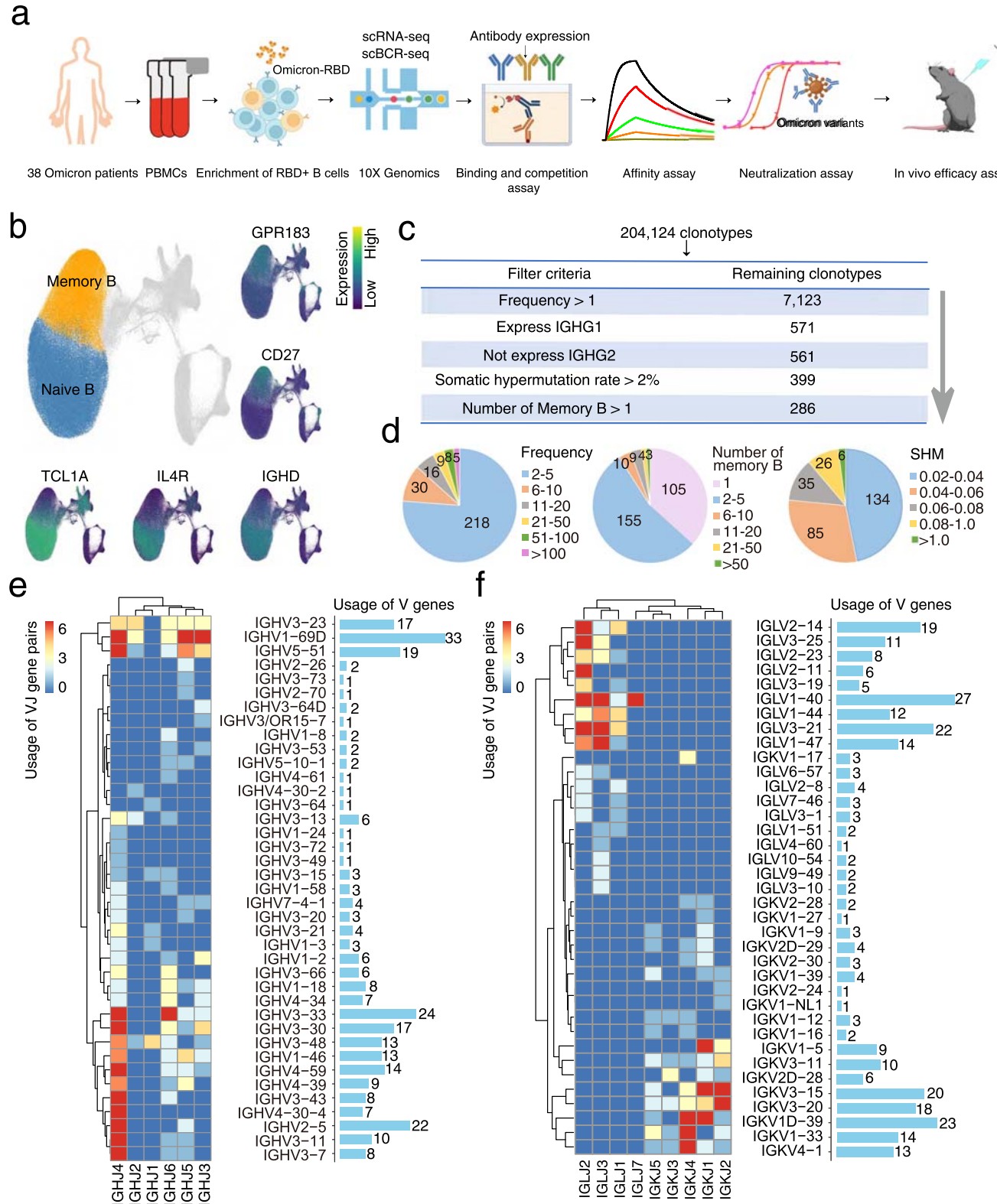

**Fig. 1 | Identification of high-confidence clonotypes by single-cell transcriptomic and BCR profiling. a** Schematic diagram of the experimental design. Single-cell RNA-seq and single-cell BCR-seq were conducted for CD19⁺ RBD⁺ B cells derived from 38 Omicron patients, followed by clonotype detection and experimental verification. **b** UMAP display of 277,630 B cells. Orange and blue correspond to memory B cells and naïve B cells, which are characterized by high expression of CD27/GPR183 and TCL1A/IL4R/IGHD, respectively. **c** Filter criteria for the selection of high-confidence clonotypes. **d** Distribution of frequency, memory B-cell number, and SHM for 286 high-confidence candidates. SHM somatic hypermutation rate. **e** VDJ gene rearrangements for heavy chains of the 286 high-confidence candidates. **f** VDJ gene rearrangements for light chains of the 286 high-confidence candidates.

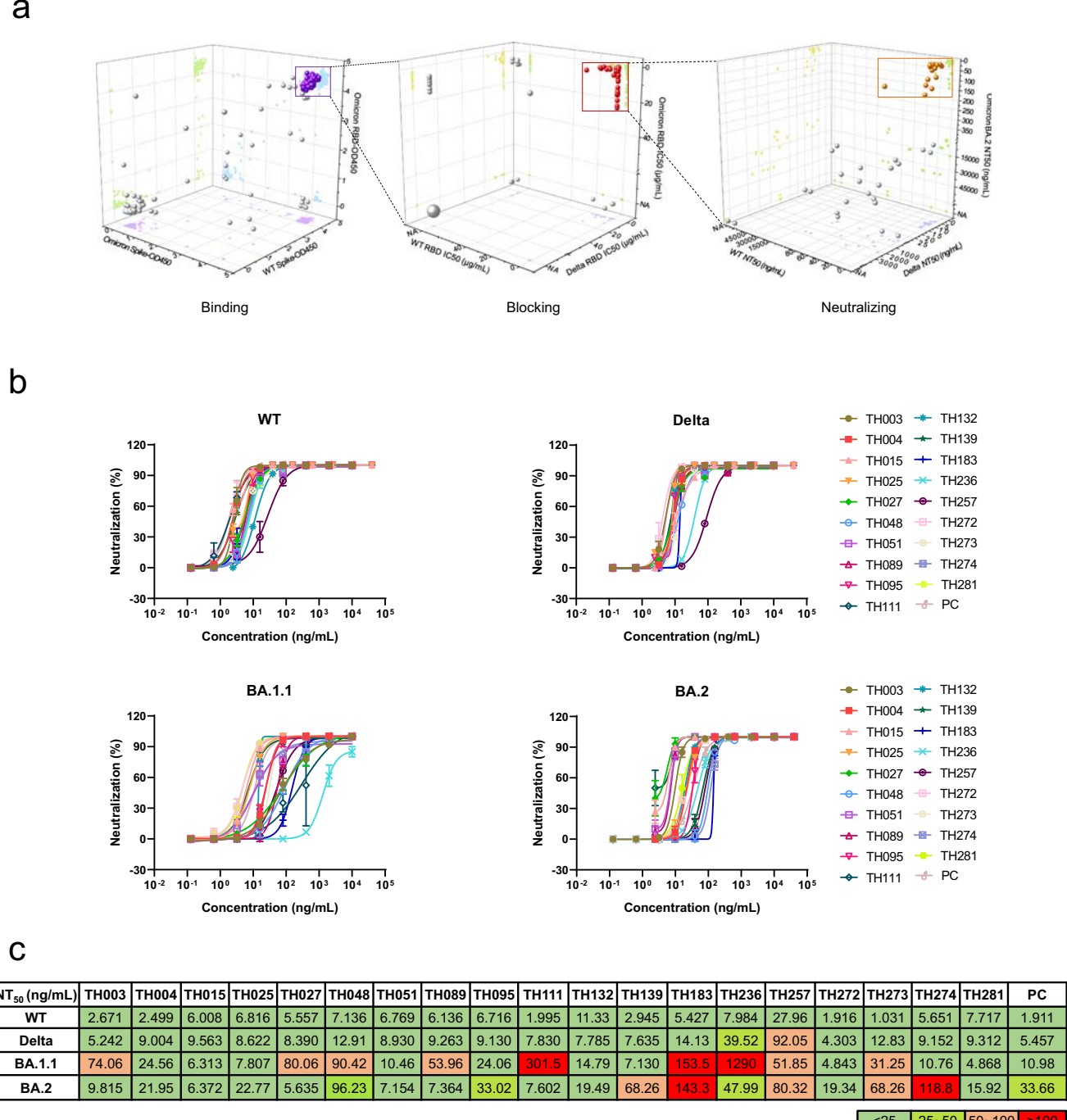

**Fig. 2 | Discovery and characterization of isolated neutralizing antibodies.**
**a** Evaluation of the binding of 286 antibodies to SARS-CoV-2 variants, measured by ELISA. The $OD_{450}$ values of each mAb against WT S protein, Omicron BA.1.1 S protein, and Omicron BA.1.1 RBD are shown. The balls in purple indicate 119 mAbs selected for further study (left). Evaluation of the blocking abilities of 119 antibodies against SARS-CoV-2 variants, measured by ELISA. The $IC_{50}$ values of each mAb against WT RBD, Delta RBD, Omicron BA.1.1 RBD are shown. The red balls refer to 44 potent neutralizing mAbs for further experiments. The ball size represents the number of antibodies. Sixty-four mAbs had no blocking ability against any variants,

represented by the large gray ball (the $IC_{50}$ values shown as NA) (middle). Evaluation of the neutralizing activity of 44 antibodies against SARS-CoV-2 variants. The $NT_{50}$ values of each mAb against the WT strain, Delta strain, and Omicron BA.2 are shown. The balls in orange indicate 19 mAbs selected for further study (right).
**b** Neutralization curves for the 19 selected antibodies against WT virus or VOCs. ($n = 3$ technical replicates). Data are presented as mean values ± SD. All experiments were performed in duplicate. **c** Neutralization potency of the 19 selected antibodies against different SARS-CoV-2 strains.

potency of 5 antibodies (TH048, TH111, TH183, TH236, and TH274) against Omicron BA.1.1 and BA.2 was significantly reduced compared with that against the Wuhan-Hu-1 and Delta strains (Fig. 2c), indicating that the virus evolved an escape profile against host adaptive immunity.

**Elite antibodies show high binding affinity and potent neutralizing activity against Omicron BA.2.12.1 and BA.4 & 5**

The 19 selected antibodies were categorized into 7 groups based on their germline origin and HCDR3 sequence (Fig. 3a). The binding affinities of 7 representative nAbs, one from each group, were analyzed by

surface plasmon resonance (SPR). Six of 7 representative nAbs displayed good binding abilities to BA.1 and BA.4/5 spike with KD values ranging from $10^{-8}$ to $10^{-13}$ M, except for TH048, which showed no binding to either the S or RBD protein of BA.4/5 (Fig. 3b and S5). The binding activity of 12 other nAbs was also screened by SPR. TH139 also almost completely lost binding affinity to both the S and RBD proteins of BA.4/5, and the other 11 nAbs showed high binding activity to various Omicron sublineages (Table S1).

We further evaluated the broad neutralization activity of 7 elite antibodies against Omicron BA.2.12.1 and BA.4/5, with neutralization potencies ranging from 0.2 to 20.7 ng/mL against Omicron BA.2.12.1 or BA.4 & 5 SARS-CoV-2-S pseudotyped virus (Fig. 3c). Only TH048 showed significantly reduced neutralization activity (more than 2 μg/mL) against Omicron BA.4/5, which was consistent with the binding affinity results. This indicates that TH048 was escaped by Omicron BA.4/5. (Fig. 3c). Six out of 7 antibodies showed potent, broadly neutralizing activity against the Omicron BA.2.12.1 and BA.4/5 strains (Fig. 3c).

## Structural studies of Omicron BA.4/5 Spike in complex with nAbs

To further analyze the molecular basis for these elite nAbs, we determined the structure of Fab in complex with Omicron BA.4/5 S-trimers by cryo-EM combined with X-ray crystallography. The cryo-EM structures of BA.4/BA.5 S-trimers in complex with 6 elite nAbs (TH003, TH027 Fab, TH132 Fab, TH236 Fab, TH272 Fab, and TH281 Fab) were obtained individually at an overall resolution of 3.4–3.7 Å (Fig. 4, S12, 13). The BA.4/5 S-trimers adopt various conformations when bound with antibodies: TH003- or TH236-Fab-bound S-trimer complexes adopt the "1-up and 2-down RBDs" conformation, while only two TH003 Fabs can be built due to structural hindrance from the neighboring RBD. TH027 and TH272 Fab-bound S-trimer complexes adopt the "2-up and 1-down" conformation, and TH132 and TH281 Fab-bound S-trimer complexes adopt the "3-up" RBD conformation (Fig. 4a).

Next, we evaluated the epitopes of these elite antibodies and found that these antibodies mainly fall into two distinct groups, which was consistent with the results of the epitope competition assay (Fig. S6). Group 1 is represented by TH003, TH027, TH236, and TH272. These antibodies, which are similar to the IGHV2–5 germline-derived LY-CoV1404[15] and XGv265[26], target the conserved solvent-exposed outer face of the SARS-CoV-2 RBD without directly blocking the binding of ACE2 and can be categorized into the RBD-5 community[2]. Group 2 antibodies, including TH132 and TH281, which target the RBM face and can be categorized in the RBD-2a community[2], directly block the binding of ACE2 with a similar binding mode to IGHV3–66/53 germline-derived F61[29], CAB-17[30], B38[31], CB6[17], BD-515[32] and ab1[33] (Fig. 4b, S9a–c).

## Structural basis for potent, pan-variant neutralizing activity

To decipher the molecular mechanism of these elite antibodies, we further analyzed the sequence and interface of the Fab-S trimer complex from two distinct groups.

Group 1 mainly originate from IGHV2–5 (TH272 and TH27), IGHV3–23 (TH003) and IGHV3–11 (TH236). TH272 tightly binds to BA.4/5 S by hydrophobic interactions and hydrogen bonds, with a buried surface area (BSA) of over 700 Å$^2$ (Table S2). L441, S443, V445, P499, T500, and Y501 of the RBD form strong hydrophobic interactions with G33, L52, W55, and I102 of the heavy chain and V29, G30, Y93, and T94 of the light chain. Extensive hydrogen bonds are formed by Y54, D56, D58, and R60 of the heavy chain and A31, Q33, Y34, and T95 of the light chain, and D56/D58 of HCDR2 form a salt bridge with K444 of the SARS-CoV-2 RBD (Fig. S7a, b). Compared to the antibodies of IGHV2–5/IGLV2–14 reported previously with an HCDR3 length of 11 amino acids and a conserved HxIxxI[27] sequence motif, the HCDR3 of TH272 is also 11 amino acids in length but contains a HxTxxT sequence motif instead. Notably, due to this difference, HCDR3 plays a weaker

role for the interaction of the TH272-S-trimer complex than previously reported IGHV2–5-origin antibodies (Fig. S8a). TH027 shares the same IGHV origin and is tightly bound to BA.4/5 S with a similar binding mode to TH272, with a longer HCDR3 loop at a length of 15 amino acids, resulting in a larger BSA of over 800 Å$^2$ (Table S2). The HCDR3 loop of TH027 forms more hydrophobic interactions and hydrogen bonds with RBD, mainly contributed by P102, N105, and Y109 with N439, L441, and V445. In addition, P107 also forms van der Waals interactions with N439, K440, and S443 (Fig. 5a, b, S8a, b). Moreover, the light chain of TH27 also contributes a larger interface and forms stronger hydrophobic interactions and salt bridges by W92 and D94 with V445, G446, and R498. This stronger interaction may explain the 5-fold higher neutralizing activity of TH027 than TH272 against the Omicron BA.4/5 pseudovirus (Fig. 3c).

Additionally, TH003 tightly binds to the BA.4/5 S-trimers by extensive hydrophobic interactions and hydrogen bonds, with a BSA of over 650 Å$^2$ (Table S2). S443, V445, P499, and T500 of the RBD form strong hydrophobic interactions with A33, Y59, and W105 of the heavy chain and G95 of the light chain. Hydrogen bonds are formed by Y53, Y59, and D104 of the heavy chain and Y32, Y34, and Y93 of the light chain. D31 of HCDR1 forms salt bridges with R346 and K444 of the RBD. N439, K440, S443, K444, G446, P499, and T500 of the RBD also form van der Waals interactions with Y32, S57, A103, and W105 of the heavy chain and D52 and Y93 of the light chain (Fig. S7c-d). Finally, TH236 originates from the IGHV3–11 germline, which has been rarely reported. TH236 contains a longer HCDR3 with 19 amino acids and forms a larger hydrogen bond network with RBD, mainly contributed by Y33, E99, P101, Y104, Y105, S107, and S108 with E340, N343, T345, R346, F347, A348, and N354 from the SARS-CoV-2 RBD (Fig. S7e, f).

Group 2 antibodies TH132 and TH281 shared the same germline origin of IGHV3–66/53 and highly identical binding epitopes. For clarity, the TH132-spike complex was used for analysis of the interaction interface. TH132 interacts with the BA.4/5 RBD with a BSA of 800 Å$^2$ (Table S2) and binds tightly to BA.4/5 S through extensive hydrophobic interactions. G416, N417, L455, F456, and Y501 of the RBD formed strong hydrophobic interactions with F52, F58, L99, and V102 of the heavy chain and P95 of the light chain. In addition, hydrogen bonds are formed by T28, R31, Y33, A53, G54, and R97 of the heavy chain and S30 and N92 of the light chain. (Fig. 5c, d, S8c, d). Notably, IGHV3–66/53 is the germline frequently used after ancestral Wuhan-Hu-1 strain infection, but the Omicron sublineages escape from most of these antibodies. In contrast, TH132 and TH281 show a broad spectrum of neutralizing activity against all Omicron sublineages. Recently, several antibodies, such as F61[29] and CAB-A17[30] from the IGHV3–66/53 germline, were also reported to show broadly potent neutralizing activity across variants. We further analyzed the amino acid sequence alignment of TH281 and TH132 with other previously reported IGHV3–66/53 antibodies (Fig. S9c) and found that several residue differences play critical roles in the cross-reactivity against different VOCs. For example, Y52F and S53A on HCDR2 form new hydrophobic interactions and hydrogen bonds with G416 and Y421 on RBD, respectively. R/Y99L on HCDR3 forms hydrophobic interactions with Y489 on RBD. Among them, G92N on LCDR3 forms a stable hydrogen bond with H505, a common mutation on all Omicron sublineages, and enhanced the interaction of antibodies. The results showed that TH132 and TH281 are new generation IGHV3–66/53 antibodies with broad and high efficiency, and epitope and sequence analysis showed that TH281 and TH132 mainly recognize conserved epitopes that are critical for hACE2 binding or RBD folding, including L455, F456, A475, Y489 and G502 (Fig. S10a). Among them, Y489 and G502 are crucial for RBD folding, and mutations of these epitopes greatly reduce the expression of RBD[34]. In addition, R403, G416, and Y421 are constrained by ACE2 binding and RBD folding, and mutations of these epitopes seriously affect ACE2 affinity and RBD expression[34]. These critical residues perform critical functions for receptor binding or RBD folding and are rarely mutated during the

## a

| Antibodies | Frequency | H-germline | L-germline | HCDR3 | MemB_count | SHM | BA.2-NT50 | Donor source | |
|---|---|---|---|---|---|---|---|---|---|
| TH003 | 184 | IGHV3-23 | IGLV2-11 | CANGVATADWYFDLW | 14 | 0.036011 | 9.815 | Bulk1, Bulk3 | ★ |
| TH004 | 170 | IGHV2-5 | IGLV2-14 | CAHMTTVTIVDYW | 98 | 0.02514 | 22.49 | Bulk1, Bulk3, Bulk2, Donor1 | |
| TH015 | 38 | IGHV2-5 | IGLV1-47 | CAHRGPGHNTPIYFDYW | 31 | 0.035135 | 6.754 | Donor1 | |
| TH025 | 19 | IGHV2-5 | IGLV2-14 | CAHHAILFVFDYW | 4 | 0.02514 | 23.40 | Bulk3 | |
| TH027 | 18 | IGHV2-5 | IGLV1-47 | CTHRGPGHNTPIYFEFW | 10 | 0.045946 | 5.610 | Donor1 | ★ |
| TH048 | 8 | IGHV3-33 | IGKV3-11 | CARDREWVSGHFDYW | 3 | 0.022161 | 100.5 | Bulk3 | ★ |
| TH051 | 7 | IGHV2-5 | IGLV1-47 | CAHRGPGHNTPIYFDYW | 5 | 0.032432 | 7.207 | Donor1 | |
| TH089 | 4 | IGHV2-5 | IGLV1-47 | CAHRGPGHNTPIYFDYW | 4 | 0.032432 | 7.369 | Bulk2, Donor1 | |
| TH095 | 4 | IGHV2-5 | IGLV2-14 | CAHKTLPTIFDSW | 3 | 0.069832 | 35.01 | Bulk1 | |
| TH111 | 3 | IGHV2-5 | IGLV1-47 | CAHRGPGHNTPIYFGYW | 2 | 0.043243 | 7.267 | Donor1 | |
| TH132 | 3 | IGHV3-66 | IGKV3-15 | CARDLGVVGATDYW | 1 | 0.025352 | 20.44 | Bulk2 | |
| TH139 | 3 | IGHV3-43 | IGKV1D-39 | CARGVYGKSHGAYGSGHIDHW | 3 | 0.07124 | 69.14 | Bulk1 | |
| TH183 | 2 | IGHV2-5 | IGLV2-14 | CAHATIPMIVGYW | 1 | 0.022346 | 144.6 | Donor1 | |
| TH236 | 2 | IGHV3-11 | IGLV2-14 | CAREQPGGYYDSSGYRLDPW | 1 | 0.047872 | 47.99 | Bulk3, Bulk2 | ★ |
| TH257 | 2 | IGHV3-43 | IGLV3-21 | CAKDMGRMKTGWHENYYMDVW | 2 | 0.044855 | 81.43 | Bulk1 | ★ |
| TH272 | 2 | IGHV2-5 | IGLV2-14 | CAHKTIPTIFDYW | 2 | 0.02514 | 19.34 | Bulk1 | ★ |
| TH273 | 2 | IGHV2-5 | IGLV2-14 | CARKGVPTIFDFW | 1 | 0.044693 | 70.58 | Bulk1 | |
| TH274 | 2 | IGHV2-5 | IGLV2-14 | CARKGVVTIFDYW | 1 | 0.039106 | 123.3 | Bulk1 | |
| TH281 | 2 | IGHV3-66 | IGKV3-15 | CARDLGVVGATDYW | 1 | 0.033803 | 15.92 | Bulk1 | ★ |

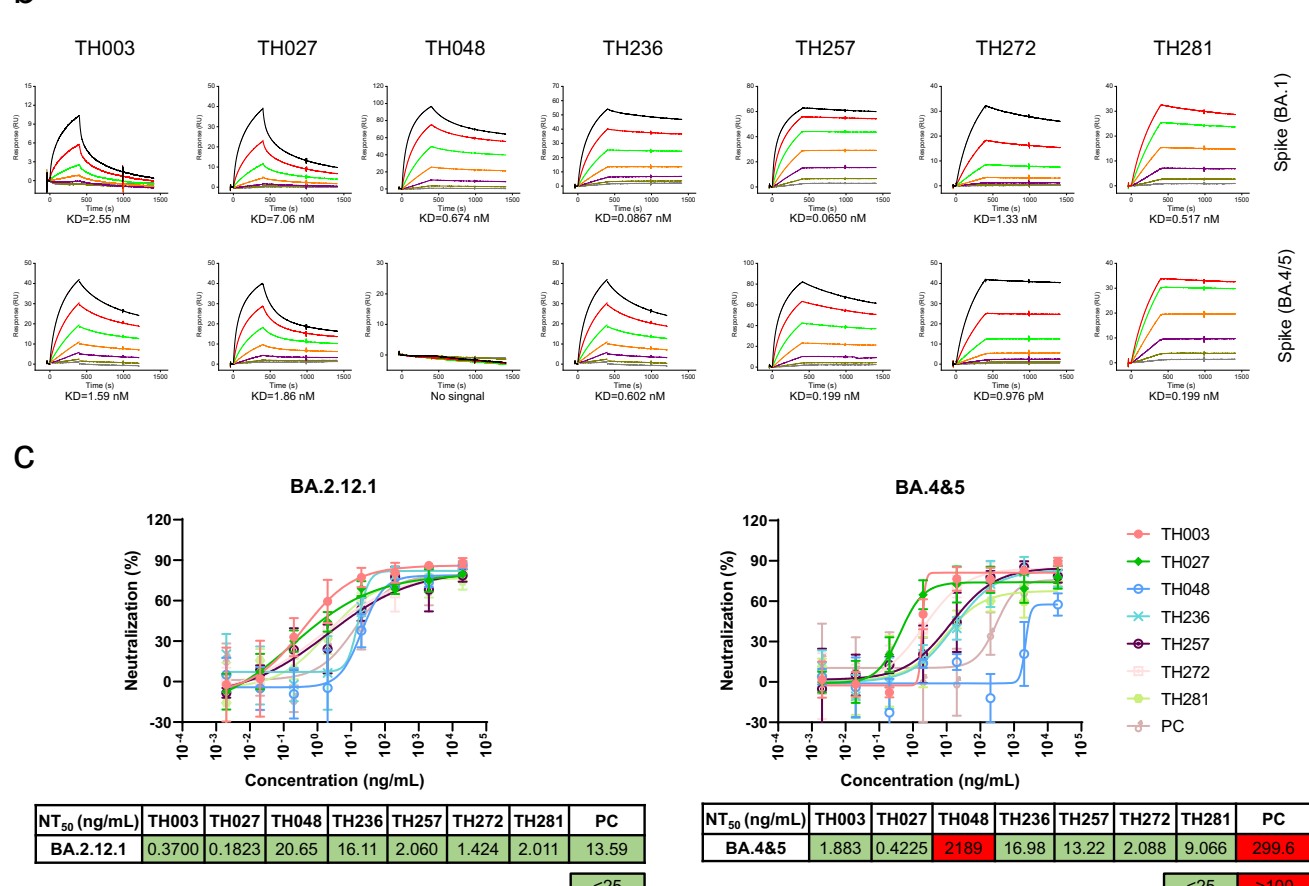

**Fig. 3 | Evaluation of elite antibodies against Omicron BA.2.12.1 and BA.4 & 5. a** Details of the 19 selected antibodies. The seven groups of antibodies are shown in different colors, and the asterisks indicate representative antibodies. **b** Binding kinetics of the seven candidate antibodies (TH003, TH027, TH048, TH236, TH257, TH272, and TH281) with the S protein of the Omicron BA.1 and BA.4/5 variants were measured by the SPR assay. **c** Omicron BA.2.12.1 or BA.4 & 5 SARS-CoV-2-S pseudotyped virus neutralization. (n = 3 technical replicates). Data are presented as mean values ± SD.

evolution of SARS-CoV-2[13,35]. Furthermore, there are two key escape mutations, N417 and Y501 (Fig. S10a), especially Y501, which has epistatic shifts and enhances hACE2 binding[35], however, these mutations do not affect the neutralizing activity of TH281 and TH132. These results suggest that TH281 and TH132 tightly bind with the critical site for hACE2 binding or RBD folding and are well tolerated against the mutation site across different VOCs.

Recently, we further evaluated the broad neutralization activity of TH027, TH132, TH272, and TH281 against emerging VOCs, such as BF.7, BQ.1.1, and XBB. The neutralizing profile showed that all these four antibodies maintain good neutralizing effects on BF.7 at single digital picomolar NT50 values. While TH027 and TH272 have been escaped by BQ.1.1 and XBB. Fortunately, TH132, TH281, and TH027 + TH132 cocktail still showed good neutralizing activity at the NT50 values of

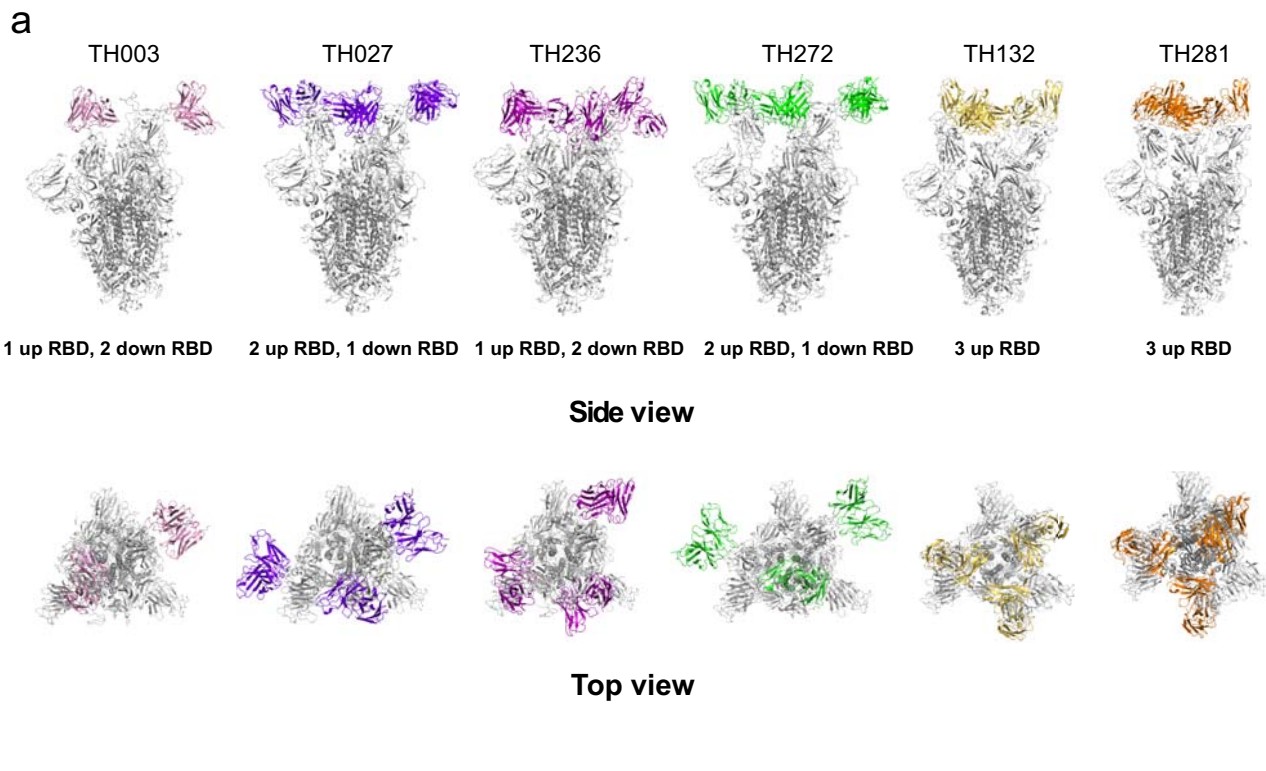

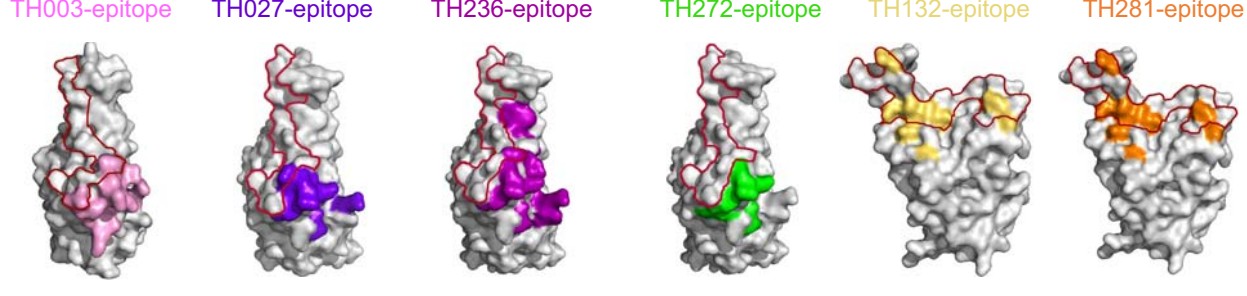

**Fig. 4 | Cryo-EM structures of the Omicron BA.4/5 spike in complex with elite nAbs. a** The structures of the BA.4/5 S trimer in complex with TH003, TH027, TH236, TH272, TH132, and TH281 are displayed in cartoon representation. TH003, TH027, TH236, TH272, TH132, and TH281 are colored pink, purple blue, purple, green, yellow, and orange, respectively, and the S trimer is colored gray. Side view (top), top view (bottom). **b** The epitopes of antibodies are displayed in surface representation. The colors are the same as in **a**, and the hACE2 interface is circled in red.

13−36 ng/mL. (Fig. 5e, f, S11a, b). Furthermore, we evaluated the binding affinity of TH027 and TH132 to the spike proteins of four variant strains: BF.7, BQ.1.1, XBB, and XBB.1.5, respectively. Both antibodies can effectively bind to the BF.7 spike, with KD values ranging from $10^{-10}$ to $10^{-11}$ M (Fig. S11c). Unfortunately, TH027 did not show any binding activity towards BQ.1.1, XBB, and XBB.1.5. In contrast, TH132 consistently exhibited a high level of affinity towards all three variants, with KD values at $10^{-9}$ M, which is consistent with the pseudovirus neutralization data (Fig. S11d, e).

### Rational pairings of noncompeting antibodies show promising therapeutic potential for nAb cocktail therapy

Previous work has shown that SARS-CoV-2 evolution is not random but constrained by a phenomenon called epistatic shifts[35], which not only maintains the high affinity for the ACE2 receptor but also enables the variants to escape from adaptive immunity and may also affect the future evolution of SARS-CoV-2[35]. One of the expected goals of our efforts was to identify elite antibodies for paired antibody cocktail therapies, with the goal of reducing the likelihood of efficacy loss against future variants and viral escape mutants that may be selected under the pressure of single antibody treatment[36]. We thus selected 7 elite antibodies for pairing in 16 two-antibody cocktails against 4 SARS-CoV-2 strains (Fig. S10b, c). Compared with a single antibody, four cocktails, TH003 + TH027, TH027 + TH132, TH027 + TH257 and TH272 + TH281, showed significant advantages against the BA.1.1 variant (Fig. 2c, S10c). To further explore the molecular mechanism of antibody cocktails, we determined the cryo-EM structures of BA.4/5 S-trimers in complex with cocktails TH027 + TH132 and TH272 + TH281 at overall resolutions of 3.6 Å and 3.8 Å, respectively (Fig. 5e, f, S12, 13). These two cocktails present similar binding modes, and both ternary complexes adopt "2-up and 1-down" RBD conformations. The 2 RBDs in the up conformation are clamped tightly by the Fabs of TH027 +

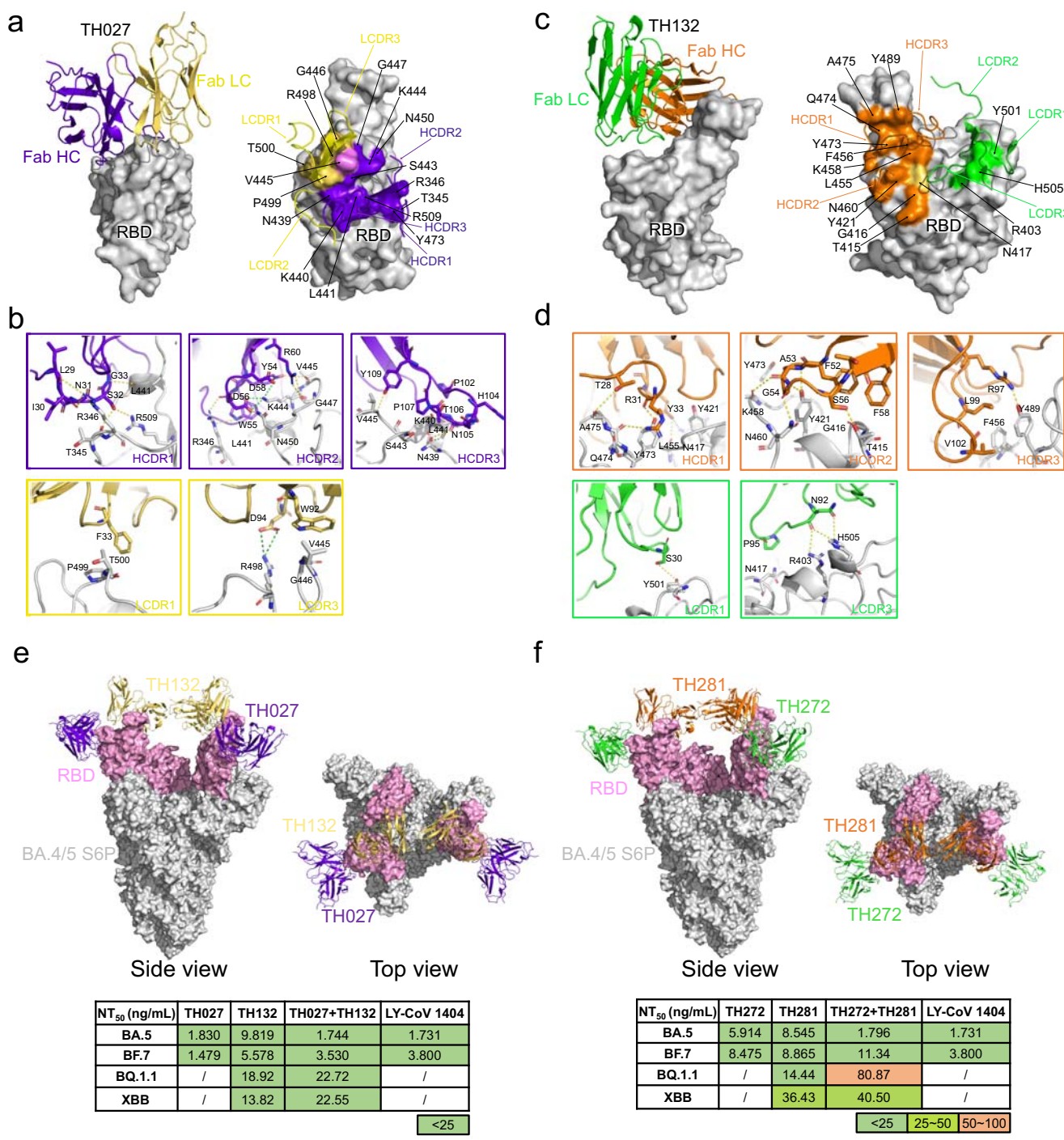

**Fig. 5 | Rational pairings of noncompeting antibodies show promising therapeutic potential for antibody therapy against emerging VOC. a** The overall structure of TH027-BA.4/5 RBD. The TH027 heavy chain (colored deep blue) and light chain (colored yellow) are displayed in cartoon representation. The BA.4/5 RBD is colored gray and displayed in surface representation (left). The epitope of TH027 is shown in surface representation. The only residue V445, which contacts both the heavy chain and light chain, is colored pink (right). **b** The interface of TH027-BA.4/5 RBD. Hydrogen bond interactions are shown as yellow dashed lines. Salt bridge interactions are shown as green dashed lines. The residues are shown in stick representation with the same colors as in **a**. **c** The overall structure of TH132-BA.4/5 RBD. The TH132 heavy chain (colored orange) and light chain (colored green) are displayed in cartoon representation. BA.4/5 RBD is colored in gray and displayed in surface representation (left). The epitope of TH132 is shown in surface representation. The only residue that contacts both the heavy chain and light chain,

N417, is colored yellow (right). **d** The interface of TH132-BA.4/5 RBD. Hydrogen bond interactions are shown as yellow dashed lines. The residues are shown in stick representation with identical colors to **c**. **e** The structure of the BA.4/5 S trimer in complex with the TH027 + TH132 cocktail, and neutralization potency of the TH027 + TH132 cocktail against different sublineages of Omicron strains. S trimer is displayed in gray surface representation. RBD is colored pink. TH027 + TH132 are displayed in cartoon representation. TH027 is colored in deep blue. TH132 is colored yellow. Side view (left). Top view (right). **f** The structure of the BA.4/5 S trimer in complex with the TH272 + TH281 cocktail, and neutralization potency of the TH272 + TH281 cocktail against different sublineages of Omicron strains. S trimer is displayed in gray surface representation. RBD is colored pink. TH272 + TH281 are displayed in cartoon representation. TH272 is colored green. TH281 is colored orange. Side view (left). Top view (right).

TH132 and TH272 + TH281. For the 1 RBD in the down conformation, the Fab molecule can hardly be built due to poor electron density, indicating potential steric hindrance from the neighboring molecule. By structural analysis, we confirmed that the pairing strategy combining RBD-2a + RBD-5 communities could tolerate the key escape mutations of BA.4/5, including L452R and F486V (Fig. S10a), and could perfectly block the outer face and RBM face of the RBD simultaneously. TH027 + TH132 and TH272 + TH281 cocktails also showed good neutralization activity against BQ.1.1 and XBB variants, and the neutralization level was at picomolar $NT_{50}$ values (Fig. 5e, f). Therefore, the ternary complex described here has potential for a new generation of antibody cocktail drugs against new emerging VOCs.

### In vivo efficacy evaluation for TH027 + TH132 cocktail

To facilitate its clinical application, the in vivo efficacy of the TH027 + TH132 cocktail against Omicron BA.5 infection was evaluated in the K18-human angiotensin-converting enzyme 2 (K18-hACE2) transgenic mouse model, which shows high susceptibility to Omicron infection. In the study, The mice were intraperitoneally injected with a single dose of TH027, TH132, or TH027 + TH132 cocktail 2 h before or after an intranasal challenge of $1 \times 10^4$ $TCID_{50}$ BA.5 virus (Fig. 6a). Administration of TH027 + TH132 cocktail significantly improved the body weight of BA.5-infected mice starting from 3 d.p.i. in both of the prophylactic and therapeutic group (Fig. 6b). Meanwhile, a single injection of 5 or 20 mg/kg of TH027, TH132, and TH027 + TH132 cocktail significantly decreased viral RNA copies in the prophylactic group compared with the vehicle group at 2 d.p.i. by 79,431, 55,757, 37,616, 76,768 fold, respectively. Therapeutic administration of the TH027 + TH132 cocktail at 5 or 20 mg/kg also effectively inhibited virus replication by 75,610 and 63,145 fold compared with the vehicle group at 2 d.p.i., respectively (Fig. 6c). Consistently, infectious viral titers in the lung tissue of antibodies-treated mice were substantially diminished compared with the vehicle group (Fig. 6d), and all treatment groups significantly decreased the virus titers below the detection limit (Fig. 6d). Meanwhile, immunofluorescence assays of lung tissue indicated that both of prophylactic and therapeutic groups significantly suppressed viral nucleocapsid protein expression comparing with the abundant expression in the vehicle group at 6 d.p.i. (Fig. 6e). Hematoxylin-eosin (HE) staining showed that the lung samples from the control group displayed severely abnormal lung tissue, irregularly arrange bronchial epithelial cells, and extensive immune infiltration in the alveoli, peri-bronchi, and peri-vessels at 6 d.p.i. In contrast, most alveolar septa and cavities were stained normally, and only small amounts of immune cell infiltration around bronchi/bronchioles and blood vessels were observed in both of the prophylactic or therapeutic groups (Fig. 6e). Therefore, these data demonstrated that TH027 + TH132 cocktail could effectively protect mice against SARS-CoV-2 infection either prophylactically or therapeutically.

### Discussion

Billions of doses of SARS-CoV-2 vaccines have been administered worldwide in the past two and a half years, and the rapid development and application of the vaccine have witnessed great scientific and medical achievements[37]. However, the protection efficacy elicited by vaccination or primary infection was severely reduced or abrogated by Omicron sublineages[4,38,39]. Moreover, the majority of the vaccines were developed based on the S glycoprotein, RBD domain, or inactivated virus of the ancestral Wuhan-Hu-1 strain, and recent work has shown that this immunological background can be counterproductive due to immune imprinting or antigen sin, so the antibodies elicited after breakthrough infection remain primarily activated against ancestral strains, leading to poor efficacy against new emerging variants[8,20,21]. This poses a severe challenge to strategies to gain herd immunity

through natural infection or vaccination and suggests that vaccine development based on emerging strains, such as Omicron sublineages BQ.1.1 and XBB, may also have difficulty achieving potent and durable protection.

In this study, by using single-cell sequencing technology, we isolated hundreds of potent neutralizing antibodies from the first Omicron breakthrough infection cohort in mainland China. The elite antibodies showed perfect neutralizing activity across variants, including the Wuhan-Hu-1, Delta, and Omicron sublineages. By using Omicron BA.1 RBD as the bait, selected antibodies were found, mainly among several highly frequently used germlines, such as IGHV2−5, IGHV3−23, and IGHV1−69D. Among them, antibodies derived from IGHV2−5, IGHV3−43, and IGHV3−66/53 constitute the majority of elite neutralizers. It is worth noting that the IGHV2−5/IGLV2−14 combination represents a public antibody response target on the SARS-CoV-2 RBD protein and shows potent pan-variant neutralizing activity. Of note, IGHV3−66/53 is one of the most frequently used germlines for the ancestral Wuhan-Hu-1 strain RBD antibody but is mostly escaped by the Omicron sublineage due to its highly antigenic drift on the RBM region. In contrast, TH132 and TH281, as reported here, show potent cross-reactivity and neutralizing activity against all the VOCs tested including BQ.1.1 and XBB. In addition, we found that these antibodies were clonally expanded and matured without significant SHM (SHM rates of 0.046 and 0.025 for TH27 and TH132, respectively). This phenomenon suggests that Omicron BA.1 breakthrough infection could also broadly induce a potent humoral response in the vaccinated population, despite the impact of antigen sin.

Here, we systematically investigate all the representative pan-variant neutralizing antibodies elicited by inactivated vaccine breakthrough infection by Omicron BA.1. TH027 and TH272 belong to the RBD-5 community derived from the IGHV2−5 germline, which engages the SARS-CoV-2 RBD at a neighboring site near the RBM. In contrast to previous work, the HCDR3 of these antibodies harbors a conserved HxTxxT motif, but not the classic HxIxxI[27] motif previously reported; consequently, the whole HCDR3 loop does not play a critical role in the antigen-antibody interaction network. Additionally, based on the amino acid sequence alignment analysis, TH027 and TH272 had D at position 54 of HCDR2, which is associated with the IGHV2−5*02 allele. These findings are in agreement with the claims of allelic preference reported by Yuan et al.[27]. Our structural insight into these antibodies suggests that the antibodies derived from the IGHV2−5/IGLV2−14 germline recognize and bind with SARS-CoV-2 RBD across variants by an extensive hydrogen bond network and hydrophobic interaction mainly constituted by HCDR1, HCDR2, LCDR1, and LCDR2, but partially by HCDR3. The antibodies derived from IGHV2−5 germline showed good potency on BF.7. Amino acid sequence alignment and structural analysis revealed that the R346T locate at the rim region of their binding epitopes, which may slightly disrupt the antibody-antigen interface but not play a critical role for this immune evasion. Moreover, TH027 and TH272 are supposed to maintain good neutralizing activity against BJ.1 and BA.4.6 due to their similarity with BF.7. However, TH027 and TH272 were significantly escaped by BQ.1.1 and XBB variants, likely due to the K444T mutation in BQ.1.1 and CH.1.1, which disrupts the salt bridge within the interface. The V445P and G446S mutations in XBB and XBB.1.5 may induce local conformational changes and create steric hindrance, leading to the immune evasion of TH027 and TH272 by XBB or XBB.1.5.

TH132 and TH281 represent the pan-variant neutralizing antibodies from the RBD-2a community derived from the well-studied IGHV3−66/53 germline. Unlike most antibodies escaped by the Omicron variant, TH132 and TH281 tightly bind with and neutralize all VOCs tested. Although their neutralizing activities against BQ.1.1 and XBB were slightly weakened, possibly due to the only mutation N460K that makes the side chain longer, creating steric hindrance with antibodies, they still retained potent neutralizing activities at

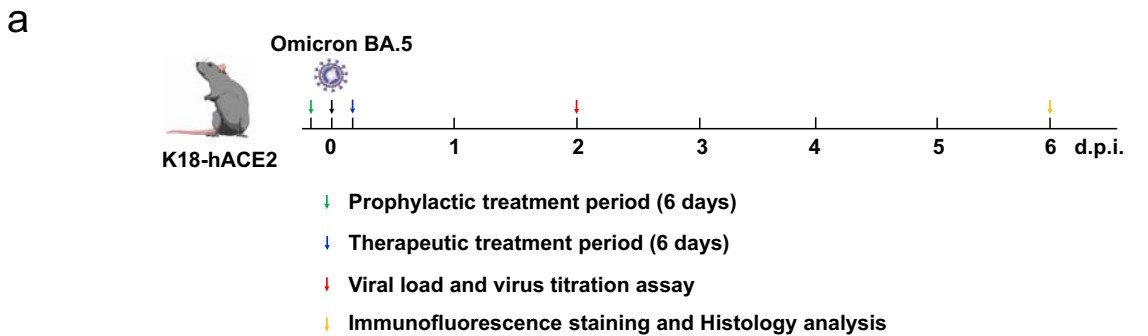

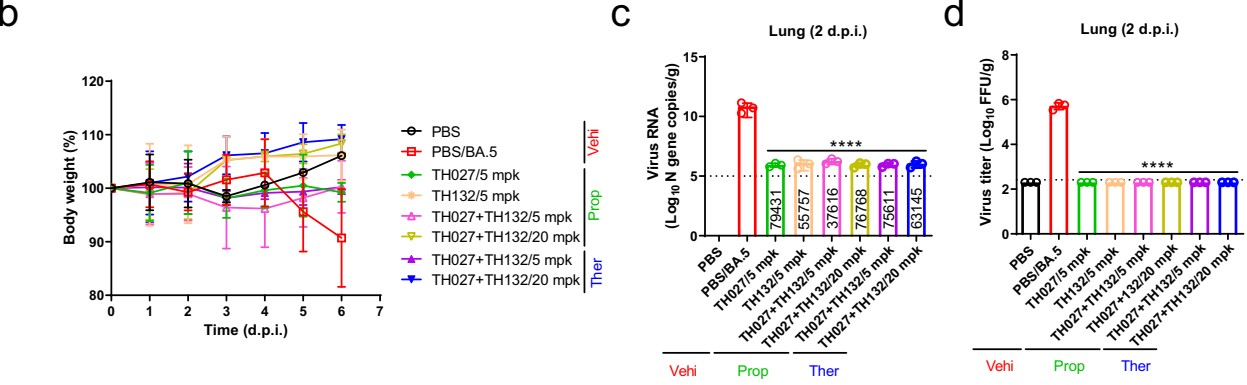

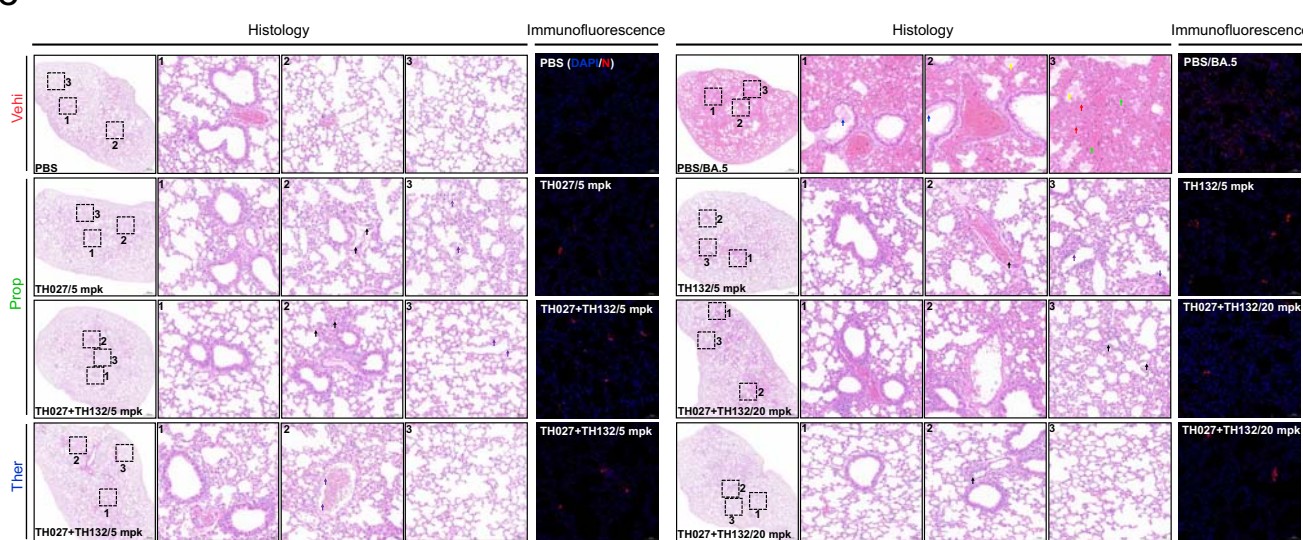

**Fig. 6 | Evaluation of in vivo efficacy of prophylactic and therapeutic TH027 + TH132 cocktail. a** Experimental design for TH027 + TH132 cocktail efficacy evaluation in K18-hACE2 transgenic mice. **b** Mice were intraperitoneally injected with a single injection of 5 or 20 mg/kg of TH027, TH132, or TH027 + TH132 cocktail 2 h before or after an intranasal challenge of $1 \times 10^4$ $TCID_{50}$ BA.5 virus. Body weight of mice were monitored daily until 6 days post-infection (d.p.i.). (PBS and TH027 + TH132/20 mpk on before 2 d.p.i.: $n = 6$ mice, other groups: $n = 5$ mice. PBS and TH027 + TH132/20 mpk on after 2 d.p.i.: $n = 3$ mice, other groups: $n = 2$ mice). **c** Viral RNA levels in the lung tissue were measured by qRT–PCR assay at 2 d.p.i. ($n = 3$ mice). **d** Viral titers in the lung tissue were measured by fluorescence focus assay at 2 d.p.i. ($n = 3$ mice). **e** Histopathological and immunofluorescence analyses of TH027, TH132, or TH027 + TH132 cocktail treated or untreated mice infected with Omicron BA.5. Representative images of lung sections stained by HE in antibodies prophylactic and therapeutic groups at 6 d.p.i. The images of bronchioles, blood vessels, and alveoli of the lungs are indicated by black dotted squares with numbers 1, 2, and 3, respectively. The images on the right (bars = 50 µm) are enlarged regions in the dashed boxes of the left images (bars = 500 µm). Red, blue, green, yellow, black, and purple arrows indicate necrotic epithelial cells in the alveoli, dead cell debris in the bronchioles, severe bleeding in the alveoli, a large amount of protein mucus in the lungs, mononuclear cell infiltrations in the blood vessels and inflammatory infiltrations in the lungs, respectively. Representative images of the *N* protein expression in the lungs at 6 d.p.i. (bars = 50 µm). Three mice were sampled in each group and 5 sections from each animal were used for histology analysis. All data are presented as mean values ± SD. Statistical differences were determined by two-way ANOVA in **c, d**. *$P < 0.05$, **$P < 0.01$, ***$P < 0.001$, ****$P < 0.0001$.

the $NT_{50}$ values of 13–36 ng/mL. Structural and sequence alignment analyses showed that TH132/TH281 form strong hydrophobic interactions and new hydrogen bonds in their HCDR2, HCDR3, and LCDR3 regions, which lead to tight binding with the critical residues for RBD folding and ACE2 binding. Based on this observation, we speculate that TH132 and TH281 would still maintain good neutralizing activity against BJ.1, BA.4.6, CH.1.1, and possibly new emerging VOCs.

Together with vaccines and small-molecule drugs, neutralizing antibodies has proven to be one of the most promising options for treating and preventing this pandemic, especially for the most vulnerable, including elderly individuals, patients with preexisting diseases, and immuno-compromised populations. Notably, some populations cannot produce antibodies using specific germlines since some elite antibodies present a strong allelic preference[27], as do TH027 and TH272 reported here. For these specific populations, neutralizing antibodies, especially in engineered forms, would provide an indispensable safeguard against infection[12]. Structural insight into these nAbs reveals the molecular basis for high affinity and potent neutralizing activity across variants. Moreover, a molecular understanding of elite antibodies also provides invaluable pairing suggestions for the development of monoclonal antibody cocktail therapies. Based on the well-characterized cryo-EM structures, we rationally paired two noncompeting antibodies in cocktails that showed exceptional neutralization breadth and potency. TH027 and TH272 do not directly block the binding of the RBM but can tightly bind with both the up and down RBDs. Meanwhile, TH132 and TH281 target the RBM face of the RBD and can bind with the RBD only in the up conformation but recognize and tightly bind with residues that are crucial for RBD folding and ACE2 binding. These structure-guided antibody cocktail pairing strategies would greatly reduce the chance of immune evasion by emerging VOCs, as proven by the complete viral clearance profile in the animal model of Omicron BA.5 infection. We anticipate that the molecular understanding of these antibodies reported here would provide a promising starting point for the development of more effective countermeasures against SARS-CoV-2 infection.

## Methods

### Sources of patients

Thirty-eight Omicron patients (20 female and 18 male patients aged 16−83 years. Table S7) in the infection stage from Tianjin Haihe Hospital were involved in this study; 34 of them received two or three doses of BBBIP-CorV inactivated SARS-CoV-2 vaccine as part of the national vaccination program before BA.1 infection, and 4 of them had not been vaccinated before infection. The volunteers' peripheral B cells were isolated and enriched for further processing. Sample collection, preprocessing, and laboratory operations were approved by the Ethics Committee of the Haihe Laboratory of Cell Ecosystem (ethical approval number: HHL2022005-EC-1). Written informed consent was obtained from each enrolled patient in accordance with the Declaration of Helsinki.

We assessed the levels of IgG antibodies specific to the Omicron RBD protein in the sera of 38 infectious Omicron patients. We observed that three patients (donor1, donor2, and donor3) produced higher titers of Omicron RBD-specific IgG antibodies, as determined by enzyme-linked immunosorbent assay. Thus, we collected 30 mL of blood from each of these three donors for library construction. For the remaining donors, we randomly divided them into three groups (bulk1, bulk2, and bulk3) and collected peripheral blood (2−5 mL from each donor) to obtain more antibodies, 60 convalescent patients' B cells were divided into six batches[24]. In total, we had 38 donors across these six groups, and we performed antigen enrichment and library construction on all of them simultaneously.

### Isolation of PBMCs from blood

All patients' blood samples were collected from Tianjin Haihe Hospital, China. Peripheral blood mononuclear cells (PBMCs) were isolated by standard density-gradient centrifugation methods using Ficoll (GE Healthcare). Parts of PBMCs were stored frozen in a cell freezing medium (90% fetal bovine serum, 10% DMSO) and thawed at 37 °C before use.

### Antigen-binding B-cell enrichment

All processing steps were conducted in a P2-level laboratory. Peripheral blood (20 ml) was collected from Omicron patients and then isolated using lymphocyte separation solution (Catalog #LTS1077, TBD) according to standard density-gradient centrifugation. Cells were harvested and resuspended in freezing media (90% FBS, 10% DMSO) and frozen using a freezing container in a −80 °C freezer for less than 20 days, which was thawed in a water bath at 37 °C before use. CD19 + B cells were obtained by the purification of human blood cells using immunomagnetic bead isolation (Catalog #17954, STEM-CELL). RBD + B cells were enriched from purified CD19 + B cells incubated with biotinylated Omicron RBD (5 µg/mL, PBS) by immunomagnetic positive selection. Then, RBD + B cells were resuspended in PBS containing 0.04% BSA. The number and viability of cells were measured using a TC20 automated cell counter (Bio-Rad). The optimal range of the cell concentration was 700−1200 cells/µL to maximize the likelihood of achieving the desired cell-recovery target.

### Single-cell 5′ mRNA and VDJ library construction and sequencing

According to the results of the cell counter and the recommended cell concentration, each sample (with -17000 single cells) was immediately loaded onto the 10× Chromium Next GEM Chip K according to the manufacturer's instructions user guide (CG000331). In detail, RBD + B cells were submitted to a Chromium single-cell 5′ mRNA library (Catalog #1000263, 10× Genomics) and V(D)J enrichment (Catalog #1000253, 10× Genomics) kits. cDNA purification and size selection were achieved by SPRI select beads (Catalog #B23318, Beckman Coulter). The quality of the cDNA post amplification, cDNA post target enrichment, and final libraries were assessed using Qubit 4.0 and Bioanalyzer 2100 with high-sensitivity chips (Catalog #5067−4626, Agilent). The libraries were sequenced using an Illumina platform Novaseq6000 with 150 bp pair-end sequencing.

### Processing of single-cell RNA-seq data

Gene-barcode matrices for 21 samples (involving 38 Omicron patients) were generated by the Cellranger (v6.1.2) "count" function referring to the human genome (GRCh38). The total gene number, total unique molecular identifier (UMI) count and percentages of mitochondrial UMIs for each cell were calculated by Seurat (v4.0.3)[40]. Scrublet (v0.2.1)[41] was used to evaluate the doublet score and identify the doublets based on UMI matrices. To cognize the subtypes of B cells, their UMI matrices were projected to the Atlas of Blood Cells (ABC) by "TransferData" from Seurat (v4.0.3). Next, UMI matrices for all samples were integrated together by Scanpy (v1.5.1)[42]. Cells were excluded if they failed to meet the following filtering criteria: *(1) > 400 genes; (2) > 800 UMIs; (3) < 15% mitochondrial UMIs, as well as excluding those doublets labeled by Scrublet (v0.2.1)[41] with default parameters. After the removal of low-quality cells, the integrated UMI matrix was normalized, logarithmized, and scaled, which was then submitted to principal component analysis (PCA) dimension reduction based on the highly variable genes calculated by the "highly_variable_genes" function from Scanpy (v1.5.1)[42] with default parameters. The BBKNN algorithm was applied to remove batch effects among different samples, followed by dimension reduction by UMAP. Finally, cell clusters of B cells were identified by the Leiden algorithm at a resolution of 0.4 and visualized by UMAP. Combining the annotations of subtypes referring to ABC with highly expressed signature genes (CD27 and GPR183), MBCs were precisely identified for subsequent clonotype selection.

### Analysis of single-cell BCR

The BCR contig sequences were assembled and annotated by the Cellranger (v6.1.2) "vdj" function against the reference genome (GRCh38). Contigs labeled as high-confidence, productive, and UMIs ≥2 were retained for clonotype constitution. Only cells with at least one

heavy chain (IGH) and one light chain (IGL or IGK) were kept for further analysis. Regarding those cells with two or more assembled IGHs or IGGs/IGKs, the dominant heavy or light chain was defined as IGHs or IGLs/IGKs with the highest UMI level. Each unique IGH-IGL/IGK pair was regarded as a clonotype, and the frequency of B cells harboring the same clonotype indicated the clonality degree of that clonotype. Among different samples and donors, clonotypes with the exactly same VDJ sequences and rearranged VDJ genes were summarized together, the frequencies of which were higher than 1, suggesting a higher chance of clonal expansion. SHM was equal to the total number of mismatches and gaps/the length of the heavy-chain VDJ DNA sequence, in which mismatches and gaps were obtained by mapping the heavy-chain VDJ DNA sequence against the germline sequence (downloaded from the international ImMunoGeneTics information system (IMGT)) using Igblast (v1.18.0)[43].

### Statistical analysis and plots

Nonparametric Wilcoxon tests were conducted by R language to compare the differences between two groups. Reported $P$ values were from two-sided tests, and a $P$ value < 0.05 was considered to be significant. Heatmaps were generated by the "pheatmap" package, while pie plots, bar plots, and boxplots were generated by the "ggplot2" package in R language. Circos plots were achieved by the "PlotFancyVJUsage" package from vdjtools (v1.2.1)[44].

### Antibody production and Fab generation

A total of 286 pairs of heavy- and light-chain plasmids were mixed with transfection reagent TF02 (Sinofection, #STF02) and added to HEK293 cells. HEK293 cells were cultured in SMS 293-SUPI medium (Sinofection, #M293-SUPI-100) at 37 °C with 5% $CO_2$. Seven days after transfection, a partial-condition medium was taken for binding identification by ELISA. Ten days after transfection, the cultured medium was harvested, and mAbs were purified using a Protein A column (GE, Hitrap Protein A HP). Antibodies use papain to generate Fab fragments. Briefly, antibodies were first cleaved by papain for 8 h at 37 °C. The mixture then attached Fc crystallizable fragments through Protein A columns, allowing Fab to flow out. The Fab was collected and dialyzed into PBS.

### ELISA binding assay

WT S protein, omicron S protein, or omicron RBD (Sino Biological Inc, #40589-V08H4; #40589-V08H26; #40592-V08H121) were coated in plates at 1 μg/mL in PBS at 4 °C overnight. After a regular washing and blocking process, 100 μL of 1 μg/mL mAb was added to each well. After a 1 h incubation at 25 °C, the plates were washed, and 100 μL of 1:5000 anti-human IgG (H + L)/HRP (Jackson, #109-035-098) was added to each well and incubated for 1 h at 25 °C. After regular washing, 100 μL of TMB (Solarbio, #PR1200) was added and incubated for 15 min. The absorbance at 450 nm was measured immediately after stop solution (Solarbio, #C1058) was added.

### ELISA blocking assay

Human ACE2 (Sino Biological Inc, #10108-H05H) was coated in plates at 2 μg/mL in PBS at 4 °C overnight. After regular washing and blocking, 100 μl 0.5 μg/mL His-tagged WT RBD (Sino Biological Inc, #40592-V08B), 0.04 μg/mL Delta RBD (Sino Biological Inc, #40592-V08H90) or 0.01 μg/mL Omicron BA.1.1 RBD (Sino Biological Inc, #40592-V08H121) was added to each well. In addition, 100 μl diluted mAbs (12 μg/mL, 4 μg/mL, 1.333 μg/mL, 0.444 μg/mL, 0.148 μg/mL, 0.0494 μg/mL, 0.0165 μg/mL) were added and mixed. After a 1 h incubation at 25 °C, the plates were washed, and 100 μL of 0.1 μg/mL anti-His-HRP (Sino Biological Inc., #A5327) was added to each well and incubated for 1 h at 25 °C. After regular washing, 100 μL of TMB (Solarbio, #PR1200) was added and incubated for 15 min. The absorbance at 450 nm was measured immediately after stop solution (Solarbio, #C1058) was added.

### Cell culture

Vero E6 (ATCC, CRL-1586) and Huh-7 (JCRB, 0403) cells were cultured in Dulbecco's modified Eagle's medium (DMEM) supplemented with 10% fetal bovine serum (FBS) and 100 IU/mL penicillin and 100 μg/mL streptomycin. All cells were cultured at 37 °C in a fully humidified atmosphere containing 5% $CO_2$ and tested negative for mycoplasma infection.

### Virus preparation and titrations

The SARS-CoV-2 wild-type strain (WT-IQTC02-16#-P5-YQ-500 μL, WT), Delta variant (Delta-IM2175251-P3-YQ-500 μL, Delta), Omicron BA.1.1 variant (Omicron BA.1.1-IM21Y6017-P4-YQ-250 μL, Omicron BA.1.1), Omicron BA.2−3 variant (Omicron BA.2−3-P2-YQ-500 μL, Omicron BA.2) and Omicron BA.5 variant (GDPCC-303-Omicron BA.5-YQ-300 μL, Omicron BA.5) were propagated in Vero E6 cells. Virus titers were determined with 10-fold serial dilutions in confluent Vero E6 cells in 96-well microtiter plates. Three days after inoculation, a cytopathic effect (CPE) was scored, and the Reed-Muench formula was used to calculate the $TCID_{50}$. SARS-CoV-2 WT, Delta, Omicron BA.1.1 and BA.2 stocks used in the experiments had undergone five, three, four, and two passages on Vero E6 cells and were stored at −80 °C, respectively. All of the infection experiments were performed at BSL-3 in Guangzhou Customs Inspection and Quarantine Technology Center (IQTC).

### Authentic SARS-CoV-2 neutralization assay

$2 \times 10^4$ Vero E6 cells were seeded in a 96-well plate for 20 h. The antibodies with different dilution concentrations were mixed with SARS-CoV-2 (MOI = 0.01) and pretreated for one hour at 37 °C, and then 200 μL mixtures were inoculated onto monolayer Vero E6 cells. Forty-eight hours after inoculation, CPE was scored by a Celigo Image Cytometer. The inhibition of antibodies and the value of half-maximal neutralizing titres ($NT_{50}$) were calculated from SARS-CoV-2's CPE rates. Three independent experiments were performed with triplicate or quadruplicate infections, and one representative is shown.

### Pseudotyped SARS-CoV-2 neutralization assay

SARS-CoV-2-S pseudotyped virus Omicron BA.2.12.1 (DR-XG-C015), Omicron BA.4 & BA.5 (DR-XG-C013), Omicron BF.7 (DR-XG-C020), Omicron BQ.1.1 (DR-XG-C021), Omicron XBB (DR-XG-C022) and XBB.1.5 (DR-XG-C031) were purchased from Guangzhou DARUI Biotechnology Co., Ltd., and the VSV-based pseudotyped SARS-CoV-2 variants were produced by transfecting 293 T cells with S protein expression plasmids and simultaneously infected with G*ΔG-VSV (Kerafast, Boston, MA). $2 \times 10^4$ Huh-7 cells were seeded in a 96-well plate. The antibodies at different dilution concentrations were mixed with SARS-CoV-2 (650 $TCID_{50}$/well) and pretreated for 1 hour at 37 °C, and then 200 μL mixtures were inoculated onto a monolayer of Huh-7 cells. Chemiluminescence signals were detected twenty-four hours after the incubation of cells and virus at 37 °C with 5% $CO_2$. The Britelite plus reporter gene assay system (PerkinElmer, Waltham, MA) and PerkinElmer Ensight luminometer were used for signal collection. The inhibition of antibodies and the value of $NT_{50}$ were calculated from luciferase expression of pseudotyped SARS-CoV-2. Two independent experiments were performed with triplicate or octuplicate infections, and one representative is shown.

### Protein expression and purification

The sequences of Omicron full-length S protein residues 13−1208 (GenBank: MN908947) and BA.4 S protein residues 1−1208 (GenBank: MN908947) were synthesized and cloned into the mammalian expression vector pCDNA3.1. Receptor-binding domain (RBD) residues 319−541 were synthesized and cloned into the baculovirus expression vector pAcGP67-A. In addition, proline substitutions at residues 817, 892, 899, 942, 986, and 987, 'GSAS' substitutions at the S1/S2 furin

cleavage site (residues 682–685), and the C-terminal T4 fibrin trimer domain, and Avi and His or Strep purification tags were introduced into the S construct to stabilize the trimeric conformation of the S protein.

The pCDNA3.1 plasmid containing the SARS-CoV-2 S protein-coding sequences was transiently transfected into Expi293F cells. The supernatant was collected, and soluble protein was purified by Strep affinity chromatography after 4 days. The S protein was further purified via gel filtration chromatography with a Superose 6 10/300 column (GE Healthcare) in a buffer composed of PBS (pH 7.4). The pAcGP67-A plasmid containing the RBD protein-coding sequences was transfected into Sf9 cells using Cellfectin II Reagent (Invitrogen). The low-titer viruses were harvested and then amplified to generate a high-titer virus stock. Viruses were coinfected with Hi5 cells. The supernatant was collected, and soluble protein was purified by Ni-NTA resin (GE Healthcare) after 72 h. The RBD protein was further purified via gel filtration chromatography with a Superdex 75 Increase 10/300 GL column (GE Healthcare). SDS−PAGE analysis revealed over 95% purity of the final purified recombinant protein.

### mAbs binding kinetics measured by SPR
The binding kinetics and affinity of mAbs to RBD/S (Sino Biological Inc.) were analyzed by SPR (Biacore 8 K, Cytiva). Specifically, purified RBDs or S proteins were covalently immobilized to CM5 sensor chips via amine groups in 10 mM sodium acetate buffer (pH 5.0). By calculating the coupling amount of RBD/S proteins, Rmax was less than 100. SPR assays were run at a flow rate of 30 mL/min in ×1 HBS-EP buffer (Catalog #BR100669, Cytiva). Serial dilutions of purified mAbs were injected, ranging in concentration from 50 to 0.068 nM. The dilution factor was 1:3. The resulting data were fit to a 1:1 binding model using Biacore Insight Evaluation Software.

### Antibody epitope competition ELISA
For competitive ELISA, antibody A was bound in a 96-well high-binding plate. Antibody B and biotinylated Omicron S protein were mixed and incubated for 1 h before being added to the plate. After 1 hr of incubation, an HRP-conjugated anti-Avi-Tag monoclonal antibody (5G11) was used to detect the Omicron S protein bound to antibody A. The percentage of inhibition was calculated.

### Electron microscopy sample preparation and data collection
For cryo-EM sample preparation of the SARS-CoV-2 S trimer in complex with TH003 Fab, TH027 Fab, TH132 Fab, TH236 Fab, TH272 Fab or TH281 Fab, 3 μL of purified BA.4/5 S at 1.5 mg/ml was mixed with 1 μL each of the Fab fragments at 1.5 mg/ml in PBS buffer solution and incubated for 30 min on ice. For cryo-EM sample preparation of BA.4/5 S trimer in complex with TH272 Fab and TH281 Fab cocktail or TH027 Fab and TH132 Fab cocktail, purified TH272 Fab fragments and TH281 Fab fragments or TH027 Fab fragments and TH132 Fab fragments were mixed with equal volumes. 1 μL Fab fragments mixture incubated with 3 μL of purified BA.4/5 S at 1.5 mg/ml for 30 min on ice. Then, 3 μL aliquot of the mixture was transferred onto a freshly glow-discharged holey carbon grid (300-mesh Quantifoil Cu R0.6/1.0). Grids were blotted for 3.0 s with blot force 0 and flash-frozen in liquid ethane cooled by liquid nitrogen using an FEI Mark IV Vitrobot (FEI) operated at 8 °C and 100% humidity. Cryo-EM data were collected on an FEI Titan Krios electron microscope operated at 300 keV with a Gatan K3 camera at ×29,000 nominal magnification in super-resolution mode and binned to a pixel size of 0.82 Å/pixel. All data were collected with a defocus range from −1.2 μm to −2.4 μm. Automated single-particle data acquisition was performed with SerialEM[45].

### Electron microscopy data analysis
A total of 2205, 2755, 7593, 2727, 2455, 4596, 4121, and 6498 micrographs of the TH003-S complex, TH027-S complex, TH132-S complex,

TH236-S complex, TH272-S complex, TH281-S complex, TH027 + TH132-S complex, and TH272 + TH281-S complex, respectively, were recorded and subjected to alignment and motion correction using MotionCorr2[46], with a six-by-five patch-based alignment. Subsequent steps were performed using cryoSPARC[47]. Then, 572,417, 917,406, 2,437,332, 944,622, 1,045,947, 1,698,839, 1,686,785 and 2,512,835 particles of the TH003-S complex, TH027-S complex, TH132-S complex, TH236-S complex, TH272-S complex, TH281-S complex, TH027 + TH132-S complex, and TH272 + TH281-S complex, respectively, were automatically selected and extracted from micrographs for reference-free 2D-classification to discard bad particles. After particle cleaning, 219,210, 519,016, 679,900, 426,739, 401,334, 397,086, 497,604, and 541,713 particles were selected and applied for ab initio reconstruction to generate 3D models as references and performed heterogeneous refinement for the TH003-S complex, TH027-S complex, TH132-S complex, TH236-S complex, TH272-S complex, TH281-S complex, TH027 + TH132-S complex, and TH272 + TH281-S complex, respectively. Afterwards, the candidate model for each complex was selected and refined using NU-Refinement to generate the final cryo-EM map for TH003-S complex, TH027-S complex, TH132-S complex, TH236-S complex, TH272-S complex, TH281-S complex, TH027 + TH132-S complex and TH272 + TH281-S complex. To improve the density quality of the interface between RBD and monoclonal antibodies, local refinement was performed for the TH027-S complex, TH132-S complex, TH236-S complex, TH272-S complex, and TH281-S complex. Local resolution ranges were also analyzed within cryoSPARC according to the gold-standard Fourier shell correlation (FSC) cut-off of 0.143[48].

### Model building and refinement
The atomic models of the complexes were generated by first fitting the chains of the SARS-CoV-2 S trimer and Fabs (PDB ID: 7XOD)[49] into the cryo-EM maps using UCSF Chimera[50]. Additionally, the model of the TH003-S complex was fitting the chain of the SARS-CoV-2 S trimer (PDB ID: 7XOD)[49] and the crystal structure of the TH003-RBD complex in this study. Fitted models were then manually rebuilt using Coot[51] according to the protein sequences and map density and real-space refined with secondary structure and geometry restraints against the map using Phenix[52]. Models were analyzed using MolProbity[53]. All the figures were generated using UCSF Chimera or PyMOL (http://www.pymol.org).

### Crystallization
The Omicron BA.4/5 RBD protein and TH003 scFv were mixed at a molar ratio of 1.5:1. The mixture was incubated on ice for 1 h and further purified by Superdex S75 (GE Healthcare). Then, 10 mg/mL RBD/scFv proteins were used for crystal screening by the vapor-diffusion sitting-drop method at 16 °C, including the Index, Crystal Screen, PEG/Ion, SaltRX from Hampton Research, and wizard I–IV from Emerald BioSystems. The rod-like diffraction crystals appeared after 2 days in the mother liquid containing 20% w/v PEG 3350 and 0.2 M potassium citrate tribasic. Crystals were dehydrated and cryo-protected in 4 M sodium formate solution at 100 K for X-ray data collection.

### X-ray data collection, processing, and structure determination
Diffraction data were collected at the Shanghai Synchrotron Radiation Facility BL10U2 (wavelength, 0.9785 Å) at 100 K. All data sets were processed using the HKL3000 package[54]. Structures were solved by molecular replacement using Phenix with the SARS-CoV-2 RBD structure (PDB ID: 7XOD)[49] and the structures of the scFv available in the PDB with the highest sequence identities. The initial model was built into the modified experimental electron density using Coot[51] and further refined in Phenix[52]. Model geometry was verified using the program MolProbity. Structural figures were drawn using the program PyMOL (http://www.pymol.org).

## Mice experiments

K18-hACE2 transgenic mice aged 6 weeks were obtained from the GemPharmatech. The use of K18-hACE2 transgenic mice has received ethical approval from the Animal Ethics Committee at Guangzhou Customs Inspection and Quarantine Technology Center (IQTC20221003). Forty-eight female hACE2 transgenic mice were divided into eight groups with six mice in each group to evaluate the efficacy of cocktail TH027 + TH132 in prophylaxis and therapy. For the prophylactic treatment group, TH027 and TH132 or cocktail TH027 + TH132 were administered at 2 h before virus challenge. For the therapeutic treatment group, cocktail TH027 + TH132 administration was delayed until 2 h.p.i. K18-hACE2 transgenic mice received an intraperitoneal dose of 5 or 20 mg/kg in a volume of 100 μL. An equivalent volume of PBS was administered as control. On the day of infection, the hACE2 mice were intranasally inoculated with either $1 \times 10^4$ TCID$_{50}$ Omicron BA.5, pre-diluted in 50 μL DMEM. Mice were killed at the designated timepoints and organ tissues were sampled for virological and histopathological analyses.

## RNA extraction and qRT–PCR

Tissue samples were lysed and extracted with the TRIzol (Invitrogen) reagent according to the manufacturer's protocols. After RNA extraction, the quantification of viral load was performed using the one-step real-time qRT–PCR and the HiScript II One-Step qRT–PCR SYBR Green Kit (Vazyme) on the Applied Biosystems QuantStudio 6 Flex. The primer and probe sequences F: 5'-GGGGAACTTCTCTTGCTAGAAT-3', R: 5'-CAGACATTTTGCTCTCAAGCTG-3', FAM-5'-TTGCTGCTGCTTGACA-GATT-3'-TAMRA.

## Fluorescence focus assay

Confluent monolayers of Vero E6 cells were incubated with 50 μL pulmonary tissue homogenate of mice with threefold serial dilutions for 2 h at 37 °C, 5% CO$_2$, in triplicate per condition. Inoculum was removed and cells were overlayed with virus growth medium containing 2% carboxymethylcellulose sodium. At 24 h post-infection, cells were fixed in 4% paraformaldeyhe and permeabilised with 0.2% Triton-X-100 and virus plaques were visualized by immunostaining using an anti-Nucleocapsid antibody, and action of HRP on a tetramethylbenzidine-based substrate.

## Histology and indirect immunofluorescence staining

Histology and indirect immunofluorescence (IF) staining were performed using established protocols[55]. In brief, the lung samples were fixed with 4% paraformaldehyde for 14 days, embedded in paraffin, and cut into 3.5-μm sections. For H&E staining, lung tissue sections were stained with Gill's hematoxylin and eosin Y (Thermo Fisher). For the IF staining, The rabbit anti-SARS-CoV-2-N immune serum (1:100) and Cy3-conjugated goat anti-rabbit IgG antibody (Proteintech) (1:1,000) were applied as primary and secondary antibodies, respectively. Cell nuclei were labeled with DAPI (Beyotime) at 1:1000 dilution. The image collections were performed by Pannoramic MIDIsystem (3DHISTECH, Budapest) and FV1200 confocal microscopy (Olympus).

## Reporting summary

Further information on research design is available in the Nature Portfolio Reporting Summary linked to this article.

## Data availability

Density maps and structure coordinates have been deposited in the Electron Microscopy Data Bank (EMDB) and the Protein Data Bank (PDB) with accession codes EMD-34181 and PDB ID 8GOU for BA.4/5 spike trimer in complex with TH003 Fab; EMD-34124 and PDB ID 7YVE for BA.4/5 spike trimer in complex with TH027 Fab; EMD-34125 and PDB ID 7YVF for BA.4/5 spike RBD in complex with TH027 Fab; EMD-34126 and PDB ID 7YVG for BA.4/5 spike trimer in complex with TH132 Fab; EMD-34127 and PDB ID 7YVH for BA.4/5 spike RBD in complex with TH132 Fab; EMD-34128 and PDB ID 7YVI for BA.4/5 spike trimer in complex with TH236 Fab; EMD-34129 and PDB ID 7YVJ for BA.4/5 spike RBD in complex with TH236 Fab; EMD-34130 and PDB ID 7YVK for BA.4/5 spike trimer in complex with TH272 Fab; EMD-34131 and PDB ID 7YVL for BA.4/5 spike RBD in complex with TH272 Fab; EMD-34133 and PDB ID 7YVN for BA.4/5 spike trimer in complex with TH281 Fab; EMD-34132 and PDB ID 7YVM for BA.4/5 spike RBD in complex with TH281 Fab; EMD-34134 and PDB ID 7YVO for BA.4/5 spike trimer in complex with TH027 + TH132 Fab; and EMD-34135 and PDB ID 7YVP for BA.4/5 spike trimer in complex with TH272 + TH281 Fab. The atomic models generated from X-ray crystallographic studies of the TH003 scFv-BA.4/5 RBD complexes have been deposited at the Protein Data Bank (PDB, http://www.rcsb.org/) under accession codes PDB 8GPY. All single-cell sequencing data have been deposited at National Genomics Data Center (https://ngdc.cncb.ac.cn/) with access number HRA003208. The single-cell sequencing data has been registered in the Human Genetic Resource Management Platform of Ministry of Science and Technology of the People's Republic of China with registration number 2023BAT0646. The single-cell sequencing data are available under restricted access for privacy protection and access can be obtained by directly contacting correspondence or application on the website (https://ngdc.cncb.ac.cn/). The access authority can be obtained for non-commercial purposes only and not their disclosure to third parties. Source data are provided with this paper.

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

## Acknowledgements

We would like to thank the staffs at Bio-Electron Microscopy Facility of ShanghaiTech University, the Advanced Bio-imaging Technology Platform of Guangzhou Laboratory for cryo-EM data collection, and BL10U2 of Shanghai Synchrotron Radiation Facility (Shanghai) for X-ray crystallography data collection, Guangzhou Laboratory (Guangzhou, Guangdong Province) for neutralization assay, and Guangzhou Customs District Technology Center (Guangzhou, Guangdong Province) for in vivo efficacy assay. This work was supported by the National Program on Key Research Project of China (2018YFE0200400 to Y. Guo., 2021YFA1100900 to T.C. and 2018YFA0507200 to X. Chen.), the National Natural Science Foundation of China (NSFC) (32271256 to Y. Guo., 81890990 and 81730006 to T.C.), The Key Program of Natural Science

Foundation of Tianjin (20JCYBJC01340 to Y. Guo.), Haihe Laboratory of Cell Ecosystem Innovation Fund (22HHXBSS00001 to T.C.), Science and Technology Project of Tianjin (22ZYJDSS00080 to Y. Guo.), the Chinese Academy of Medical Sciences (CAMS) Innovation Fund for Medical Sciences (2021-I2M-1-040 to T.C.), the Non-CAMS Fundamental Research Funds for Central Research Institutes (3332021093 to T.C.), Basic and Applied Basic Research Projects of Guangzhou Basic Research Program (2023A04J0161 to Q.Y.), Emergency Key Program of Guangzhou Laboratory (EKPGL2021008 to Y. Guo.), R&D Program of Guangzhou Laboratory (SRPG22-002 to Z. Sun, SRPG22-003 to Z.R.).

## Author contributions

Y. Guo., Z.R., T.C., P.Z., F.L., D.X., X. Chen., X.Y. conceived the project. Y.L., X. Cheng., Jia Zhou, J.Q., L.L., S.M., B.Z., Q. H. collected the convalescent PBMC, serum and constructed VDJ libraries. X.W., Y.X., J.W. are responsible for human-material-related work. X.X., P.Z., S.Z., F.D. performed bioinformatics analysis. G.Z., Y.L., Yu Gao., H.G., H.Y., H.S. performed gene construction, protein expression, and purification. F.L., J.L., G.Z., Yan.Gao., Z.L. performed the collection of Cryo-EM data and structure determination. W.W., Yu.Gao., Z.Song. performed the collection of X-ray data and structure determination. Y.L., X.Cheng., G.Z. performed the in vitro binding assays, epitope competitive assays, and analysis data. Q.Y., J.T., G.Z., Z. Sun, H.W., X.Y. performed authentic or pseudovirus neutralization assays. Q.Y., J.D., Y.S., G.Z. performed animal experiments and analyzed the results. Y.Guo., G.Z., Q.Y., T.C., D.X., F.L., Jun Zhou., H.W., and X. Chen designed the experiments and wrote the manuscript. All authors read and approved the contents of the manuscript.

## Competing interests

T.C., Z.R., Y. Guo., P.Z., D.X., X. Chen., X.W., Y.L., X. Cheng., G.Z., X.X., F.L., Q.Y., Z.L., J.T., and J.L. declare the following competing interests: patent has been filed for some of the antibodies presented here (patent application number: 202211015148.3). All other authors declare no competing interests.

## Additional information

[1]State Key Laboratory of Medicinal Chemical Biology and College of Life Sciences, Nankai University, 38 Tongyan Road, Tianjin 300071, China. [2]State Key Laboratory of Experimental Hematology, National Clinical Research Center for Blood Diseases, Haihe Laboratory of Cell Ecosystem, Institute of Hematology & Blood Diseases Hospital, Chinese Academy of Medical Sciences & Peking Union Medical College, Tianjin 300020, China. [3]Guangzhou Laboratory, Guangzhou, Guangdong, People's Republic of China. [4]Beijing Institute of Biological Products Company Limited, China National Biotech Group, Beijing 100176, China. [5]CNBG-Nankai Joint Research Center, Nankai University, 94 Weijin Road, Tianjin 300071, China. [6]Frontiers Science Center for Cell Responses, Nankai University, 94 Weijin Road, Tianjin 300071, China. [7]Shanghai Institute for Advanced Immunochemical Studies, ShanghaiTech University, Shanghai 201210, P.R. China. [8]Guangzhou Customs District Technology Center, Guangzhou 510700, China. [9]Tianjin Haihe Hospital, Jingu Road, Tianjin 300071, China. [10]These authors contributed equally: Yu Guo, Guangshun Zhang, Qi Yang, Xiaowei Xie, Yang Lu, Xuelian Cheng, Hui Wang, Jingxi Liang, Jielin Tang, Yuxin Gao. ✉e-mail: guoyu@nankai.edu.cn; yang_qi@gzlab.ac.cn; yangxiaoming@sinopharm.com; dsxiong@ihcams.ac.cn; liu_fengjiang@gzlab.ac.cn; chen_xinwen@gzlab.ac.cn; zhuping@ihcams.ac.cn; wangximo@126.com; chengtao@ihcams.ac.cn; raozh@mail.tsinghua.edu.cn

