## [Peer Review File · Nature Communications]

REVIEWER COMMENTS

Reviewer #1 (Remarks to the Author):

Guo et al. describe the isolation of monoclonal antibodies from Omicron BA.1 breakthrough infections starting from 38 individuals, a subset of whom were previously vaccinated with an inactivated SARS-CoV-2 vaccine based on the ancestral strain. They use 10X Genomics to sequence antibody heavy and light chains from B cells sorted with a biotinylated Omicron BA.1 RBD probe and identify 204,124 clonotypes with distinct VDJ sequences.

From this, they narrow down the number of antibodies to express to 286 clones, which they divide into 7 groups based on genetic features of the antibody heavy and light chain sequences. They identify 19 neutralizing antibodies and focus on those using the IGHV2-5 germline gene, which were over-represented in the dataset, as well as antibodies using the IGHV3-66 germline gene, the latter which represent a well-described class of RBD antibodies.

The paper is well-written but has the major weakness that data on neutralizing activity against the more recently emerged Omicron sub-lineages are lacking. At this point of the pandemic such data are needed to support the claim that ultrapotent pan-variant SARS-CoV-2 neutralizing antibodies have been isolated. Therefore, this should be added and if there is substantial reduction in efficacy of the monoclonal antibodies isolated here against these variants, this should be shown, and the claims should be adjusted accordingly. Variants to test include those encoding a mutation in RBD position 356 such as BA.275.2 and BA4.6, and variants such as XAW, BJ.1, BQ1.1 and XBB.

Additional concerns and questions are summarized here:

- Did the authors mix the cells from the 38 donors or is the donor source identifiable in the data? This is necessary to know to understand which donors the final set of antibodies were isolated from.
- The IGHV gene usage plot in Figure 1e shows that the most frequently used IGHV gene was IGHV1-69D while IGHV1-69 was not used at all, which is surprising. Given that the nucleotide sequence of IGHV1-69D and IGHV1-69*01 are identical it is hard to understand how the authors know that the 33 antibodies referred to in the bar graph in Figure 1e use IGHV1-69D rather than IGHV1-69*01. The authors should explain this and if needed correct the Figure.
- The authors identified 19 neutralizing mAbs (which is not many from 38 donors) and divided these into 7 groups. How were these defined, were all antibodies assigned to a given group clonally related to each other? Were some antibodies within a group public, i.e. identified in multiple independent donors? This information should be added to the table in Figure 3a.

- Line 94: The authors write, “The discovery and molecular understanding...” and reference Yuan et al. (ref 17), which does not seem to be an appropriate reference for such a general statement. Yuan et al. (ref 17) should instead be referred to when IGHV2-5-using antibodies are mentioned the first time. In fact, the authors of the current manuscript do not mention the allele bias described in Yuan et al. showing the IGHV2-5*02 alleles that have a D in position 54 of the HCDR2 as opposed to an N in IGHV2-5*01. The authors should comment on this finding and state if variation (a D or an N) in position is expected to impact binding of the TH027 and the related IGHV2-5-using antibodies isolated in the current study. Which allele were the IGHV2-5-using antibodies isolated here assigned to? If they were assigned to the *02 allele it would strengthen the claims of Yuan et al. If they were assigned to the *01 allele this should be shown and discussed in relation to the work by Yuan et al.
- The data shown in Fig. S4a-c should be replotted as it is not easily interpreted in its present form.
- As described in Figure 1c, the authors selected antibody sequences that had over 2% SHM. However, they do not discuss whether these antibodies were de novo elicited by the breakthrough infection or if they were already present from the prior vaccination and therefore recalled. This is an interesting point that should be addressed, ideally experimentally.
- On line 413, the authors write “...some elite antibodies present strong allelic preference” with a reference to Vanshylla et al. (ref 35). This does not appear to be the correct reference for this statement. Please explain.
- An additional issue with referencing relates to statements on line 98 and line 365, where the authors discuss immune imprinting. Reference 18 should be removed in both places as this is not a primary paper but a commentary.

Reviewer #2 (Remarks to the Author):

In their manuscript, Guo et al investigate the antibody response in patients that had received multiple doses of the inactivated SARS-CoV-2 vaccine (BBBIP-CorV - Sinopharm, commonly used in China) after BA.1 breakthrough. They employ single cell sorting in combination with biophysical and EM/crystallography testing to identify broadly neutralising antibodies, and further develop them into antibody cocktails which they test in the K18-hACE2 model.

This is an interesting and well executed study which is suitable for publication in Nature Communications.

I have two minor comments:

1) Please expand the discussion, and in particular discuss how the use of the inactivated Sinovac vaccine may have affected the outcome of the study in comparison to the globally more widely used mRNA vaccines.

2) Please discuss, based on the reported mutations in XXB.1 and XXB.15, if the antibodies and cocktails here might also be applicable to these current VOCs. This could be addressed through structural modelling, or alternatively through RBD binding or viral assays.

Minor technical comment:

Line 421 "binding with the RBD" should presumably be "binding of the RBM (to ACE2)".

Reviewer #3 (Remarks to the Author):

The continued antigenic evolution of SARS-CoV-2 has led to numerous Omicron sublineages, which continue to drive waves of infection. These variants have escaped neutralization by all clinical-stage monoclonal antibodies (mAbs) due to spike protein receptor-binding domain mutations. Until recently, two FDA-approved mAb therapies (Bebtelovimab and Evusheld) were still available to treat individuals at high risk of progressing to severe COVID-19. However, recent data suggest that these are no longer effective against the most recent variants, such as XBB and BQ.1.1 (1). As such, there is a continued need to develop therapeutic monoclonal antibodies to combat SARS-CoV-2 infections. The manuscript by Guo et al describes several mAbs, obtained from BA.1 breakthrough-infected patients, capable of neutralizing D614G, Delta and several Omicron sublineages such as BA.2.12.1 and BA.4/5. This is a well-written, comprehensive study, but given the routine antibody discovery pipeline and highly competitive/fast-paced field of anti-SARS-CoV-2 antibodies, this paper does not offer much in terms of originality. Moreover, the manuscript is outdated with respect to the current pandemic situation. Conceivably, the mAbs described here could help to address the urgent clinical need for therapeutics to prevent or treat COVID-19. Still, their efficacy against more relevant Omicron variants needs to be demonstrated.

Main points to address

1. The ability to neutralize BA.2.12.1 and BA.5 is of diminishing relevance to the current pandemic situation. BQ.1.1, BA.4.6.3, XBB, and CH.1.1 can all escape neutralization by bebtelovimab, which also worked well against BA.4/5. Experimental evidence for efficacy against these VOCs should be performed by live or pseudovirus neutralization assays. If any of these molecules can differentiate themselves from the long list of ineffective mAbs, it would demonstrate the relevance of the author's work and potential clinical application.

2. Related to point one, if any of these mAbs can still neutralize currently circulating VOCs, the paper will require a rewrite to better place the results in the context of the current pandemic situation.

Reference

1) Cao et al, 2022: (<https://www.nature.com/articles/s41586-022-05644-7>)

Point-to-point response to reviewers' comments

Reviewer #1 (Remarks to the Author):

Guo et al. describe the isolation of monoclonal antibodies from Omicron BA.1 breakthrough infections starting from 38 individuals, a subset of whom were previously vaccinated with an inactivated SARS-CoV-2 vaccine based on the ancestral strain. They use 10X Genomics to sequence antibody heavy and light chains from B cells sorted with a biotinylated Omicron BA.1 RBD probe and identify 204,124 clonotypes with distinct VDJ sequences.

From this, they narrow down the number of antibodies to express to 286 clones, which they divide into 7 groups based on genetic features of the antibody heavy and light chain sequences. They identify 19 neutralizing antibodies and focus on those using the IGHV2-5 germline gene, which were over-represented in the dataset, as well as antibodies using the IGHV3-66 germline gene, the latter which represent a well-described class of RBD antibodies.

The paper is well-written but has the major weakness that data on neutralizing activity against the more recently emerged Omicron sub-lineages are lacking. At this point of the pandemic such data are needed to support the claim that ultrapotent pan-variant SARS-CoV-2 neutralizing antibodies have been isolated. Therefore, this should be added and if there is substantial reduction in efficacy of the monoclonal antibodies isolated here against these variants, this should be shown, and the claims should be adjusted accordingly. Variants to test include those encoding a mutation in RBD position 356 such as BA.275.2 and BA4.6, and variants such as XAW, BJ.1, BQ1.1 and XBB.

Response: We sincerely appreciate the reviewer for the positive evaluation of our work and raising this critical question. In fact, most of the neutralization assay and cryo-EM structure study of the reported antibodies in this paper were completed in June 2022, prior to the emergence of new VOCs such as BF.7, BQ.1, and XBB. We are delighted to observe that the selected antibody pairs using the strategies described in this manuscript have maintained potent neutralization capacity against the emerging VOCs. Currently, a cocktail of one selected antibody pair (TH27+TH132), after undergoing antibody engineering for prolonged half-life, is under CMC development and safety evaluation, and a phase I clinical trial is scheduled to be initiated in mid-2023. We hope that this cocktail can provide relatively long-term protection for the elderly, immunocompromised populations, or individuals with pre-existing conditions. Therefore, we pay close attention to the neutralizing capacity of these antibodies against the emerging VOCs.

In this revision, we provided more data on neutralizing activities of TH027, TH132, TH272 and TH281 against emerging VOCs, especially the variants encoding R346T mutation such as BF.7, BQ1.1, and XBB. The neutralizing profile showed that all these four antibodies maintain good neutralizing effect on BF.7 at single digital picomolar NT₅₀ values. While TH027 and TH272 have been escaped by BQ.1.1 and XBB. Fortunately, TH132, TH281, and TH027+TH132 cocktail still showed good neutralizing activity at the NT₅₀ values of 13-36 ng/mL.

Accordingly, we provided the neutralization data as figure.S11 and updated figure.5e/f as follow:

Figure1. updated neutralizing profile of selected antibodies on VOCs

Moreover, to better explaining the molecular basis of the antibody evasion profile, we also update figure.S10 as follow:

Figure2. updated analysis of binding epitopes on emerging VOCs

The antibodies derived from IGHV2-5 germline (TH027, TH272, and LY-CoV1404) showed good potency on BF.7. Amino acid sequence alignment and structural analysis revealed that the R346T locate at the rim region of their binding epitopes, which may slightly disrupts the antibody-antigen interface but not play critical role for this immune evasion. Moreover, TH027 and TH272 are

supposed to maintain good neutralizing activity against BJ.1 and BA.4.6 due to their similarity with BF.7. However, TH027 and TH272 were significantly escaped by BQ.1.1 and XBB variants, likely due to the K444T mutation in BQ.1.1 and CH.1.1, which disrupts the salt bridge within the interface. The V445P and G446S mutations in XBB and XBB.1.5 may induce local conformational changes and create steric hindrance, leading to the immune evasion of TH027 and TH272 by XBB or XBB.1.5.

In contrast, TH132 and TH281, derived from the IGHV3-53/66 germline and categorized in the RBD-2a community, tightly bind and neutralize all tested VOCs, including BF.7, BQ.1.1, and XBB. Although their neutralizing activities against BQ.1.1 and XBB were slightly weakened due to the N460K mutation, they still retained potent neutralizing activities at the NT₅₀ values of 13-36 ng/mL. Structural and sequence alignment analyses revealed that TH132/TH281 bind tightly with the critical residues for RBD folding and ACE2 binding. Based on this observation, we speculate that TH132 and TH281 would maintain good neutralizing activity against BJ.1, BA.4.6, CH.1.1, and possibly new emerging VOCs.

Based on the above analysis, we rewrote our discussion section accordingly.

Over the past three years, virologists and pharmacologists have been diligently exploring methods for predicting viral evolution and immune escape to guide and optimize antibody selection strategies. Despite the rapid development of many mAb drugs, they continue to face the challenge of emerging VOCs. Studies have shown that several antibodies, such as LY-CoV1404 (bebtelovimab), COV2-2196+COV2-2130 (Evusheld), and REGN10933+REGN10987 (REGN-COV2), have exhibited varying degrees of immune escape against emerging VOCs, particularly BF.7, BQ.1, and XBB. These findings highlight the challenges and limitations in antiviral antibody therapies. Nevertheless, the process of confronting these challenges has deepened our understanding of viral evolution and immune escape.

Fortunately, in our study, several antibodies that we selected in June 2022 have maintained their broad-spectrum potency against most VOCs. This data confirms that the antibody selection strategy we reported in this manuscript is effective, and we are excited to share our progress with peers and contribute to the research on antiviral therapy.

Additional concerns and questions are summarized here:

- Did the authors mix the cells from the 38 donors or is the donor source identifiable in the data? This is necessary to know to understand which donors the final set of antibodies were isolated from.

Response: Prior to conducting antigen-enriched single-cell RNA-seq, we first assessed the levels of IgG antibodies specific to the Omicron RBD protein in the sera of patients. We observed that three patients (donor1, donor2, and donor3) produced higher titers of Omicron BA.1 RBD-specific IgG antibodies, as determined by enzyme-linked immunosorbent assay. Thus, we collected 30 ml blood from each of these three donors for library construction. For the remaining donors, we randomly divided them into three groups (bulk1, bulk2, and bulk3) and collected peripheral blood (2-5 mL from each donor) to obtain more antibodies, as previously described in which 60 convalescent patients' B cells were divided into six batches (citation: Cao et al. Potent Neutralizing Antibodies against SARS-CoV-2 Identified by High-Throughput Single-Cell Sequencing of Convalescent Patients' B Cells. doi:10.1016/j.cell.2020.05.025.). In total, we had 38 donors across these six groups, and we performed antigen enrichment and library construction on all of them simultaneously.

We have rewritten some description in the Material and Method section, included the final set of 286 antibodies, along with their corresponding donor sources, as source data in the revision.

- The IGHV gene usage plot in Figure 1e shows that the most frequently used IGHV gene was IGHV1-69D while IGHV1-69 was not used at all, which is surprising. Given that the nucleotide sequence of IGHV1-69D and IGHV1-69*01 are identical it is hard to understand how the authors know that the 33 antibodies referred to in the bar graph in Figure 1e use IGHV1-69D rather than IGHV1-69*01. The authors should explain this and if needed correct the Figure.

Response: We would like to express our gratitude to the reviewer for bringing this issue to our attention. According to the Ensembl database, the base and amino acid sequences of IGHV1-69 and IGHV1-69D are highly similar, suggesting that their structures and functions are likely to be analogous. However, they differ in their genomic locations and some sequence mismatches, as illustrated in Figure 1. During the contig alignment process using Cell Ranger, the software aligns the sequences to the corresponding (BCR or TCR) reference sequences based on 12-mer perfect matches, and outputs the most accurate alignments for downstream statistical analysis.

b

IGHV1-69 AGCATCACATAACAACCACATTCCTCCTCTGAAGAAGCCCCTGGGAGCACAGCTCATCAC
 IGHV1-69D AGCATCACATAACAACCACATTCCTCCTCTGAAGAAGCCCCTGGGAGCACAGCTCATCAC
 IGHV1-69 MGGI I P I FGTANYAQKFQGRVTITALEKSTSTAYMELSSLRSEDTAVYYCAR
 IGHV1-69D MGGI I P I FGTANYAQKFQGRVTITALEKSTSTAYMELSSLRSEDTAVYYCAR

Figure 3. Alignment of IGHV1-69 and IGHV1-69D genes

The figure above shows the differences between IGHV1-69 and IGHV1-69D genes. These two genes have different genomic locations, with IGHV1-69 located at chr14:106,714,682-106,715,181 (500 bp) and IGHV1-69D located at chr14:106,762,092-106,762,588 (497 bp) in the GRCh38/hg38 database.

As a result, we presented IGHV1-69D instead of IGHV1-69 in the bar graph, as this result is the directly output from the Cell Ranger software. We prefer keep this figure without correction with the permission of the reviewer.

- The authors identified 19 neutralizing mAbs (which is not many from 38 donors) and divided these into 7 groups. How were these defined, were all antibodies assigned to a given group clonally related to each other? Were some antibodies within a group public, i.e. identified in multiple independent donors? This information should be added to the table in Figure 3a.

Response: Thank you for raising this important concern. We divided 19 elite antibodies into 7 groups mainly according to their germline origin and HCDR3 sequence. We provided more detail in Materials and Method section and updated Figure3a accordingly.

a

Antibodies	Frequency	H-germline	L-germline	HCDR3	MemB_count	SHM	BA.2-NT50	Source
TH003	184	IGHV3-23	IGLV2-11	CANGVATADWYFDLW	14	0.036011	9.815	Bulk1, Bulk3
TH004	170	IGHV2-5	IGLV2-14	CAHMTTVTIVDYW	98	0.02514	22.49	Bulk1, Bulk3, Bulk2, Donor1
TH015	38	IGHV2-5	IGLV1-47	CAHRGPGHNTPIYFDYW	31	0.035135	6.754	Donor1
TH025	19	IGHV2-5	IGLV2-14	CAHHAILFVFDYW	4	0.02514	23.40	Bulk3
TH027	18	IGHV2-5	IGLV1-47	CTHRGPGHNTPIYFEFW	10	0.045946	5.610	Donor1
TH048	8	IGHV3-33	IGKV3-11	CARDREWWSGHFDYW	3	0.022161	100.5	Bulk3
TH051	7	IGHV2-5	IGLV1-47	CAHRGPGHNTPIYFDYW	5	0.032432	7.207	Donor1
TH089	4	IGHV2-5	IGLV1-47	CAHRGPGHNTPIYFDYW	4	0.032432	7.369	Bulk2, Donor1
TH095	4	IGHV2-5	IGLV2-14	CAHKTLPITFDWS	3	0.069832	35.01	Bulk1
TH111	3	IGHV2-5	IGLV1-47	CAHRGPGHNTPIYFGYW	2	0.043243	7.267	Donor1
TH132	3	IGHV3-66	IGKV3-15	CARDLGVVGATDYW	1	0.025352	20.44	Bulk2
TH139	3	IGHV3-43	IGKV1D-39	CARGVYKSHGAYGSGHIDHW	3	0.07124	69.14	Bulk1
TH183	2	IGHV2-5	IGLV2-14	CAHATIPMIVGYW	1	0.022346	144.6	Donor1
TH236	2	IGHV3-11	IGLV2-14	CAREQPGGYDSSGYRLDPW	1	0.047872	47.99	Bulk3, Bulk2
TH257	2	IGHV3-43	IGLV3-21	CAKDMGRMKTGWHENYMDVW	2	0.044855	81.43	Bulk1
TH272	2	IGHV2-5	IGLV2-14	CAHKTIPITFDYW	2	0.02514	19.34	Bulk1
TH273	2	IGHV2-5	IGLV2-14	CARKGVPTIFDFW	1	0.044693	70.58	Bulk1
TH274	2	IGHV2-5	IGLV2-14	CARKGVITFDYW	1	0.039106	123.3	Bulk1
TH281	2	IGHV3-66	IGKV3-15	CARDLGVVGATDYW	1	0.033803	15.92	Bulk1

Figure 4. Information of the 19 selected antibodies.

- Line 94: The authors write, “The discovery and molecular understanding...” and reference Yuan et al. (ref 17), which does not seem to be an appropriate reference for such a general statement. Yuan et al. (ref 17) should instead be referred to when IGHV2-5-using antibodies are mentioned the first time. In fact, the authors of the current manuscript do not mention the allele bias described in Yuan et al. showing the IGHV2-5*02 alleles that have a D in position 54 of the HCDR2 as opposed to an N in IGHV2-5*01. The authors should comment on this finding and state if variation (a D or an N) in position is expected to impact binding of the TH027 and the related IGHV2-5-using antibodies isolated in the current study. Which allele were the IGHV2-5-using antibodies isolated here assigned to? If they were assigned to the *02 allele it would strengthen the claims of Yuan et al. If they were assigned to the *01 allele this should be shown and discussed in relation to the work by Yuan et al.

Response: Thank you for this important concern. We updated this citation accordingly. Based on the amino acid sequence alignment analysis, TH027 and TH272 both had D at position 54 of HCDR2, which is associated with the IGHV2-5*02 allele. These findings are in agreement with the claims made by Yuan et al. As a result, we have updated the discussion section accordingly to reflect these findings.

- The data shown in Fig. S4a-c should be replotted as it is not easily interpreted in its present form.

Response: We thank this reviewer for pointing this out. As per their recommendation, we have transformed the dot plots into box plots, enabling us to better comprehend the distinctive features of the 19 optimized clonotypes (refer to Fig. 4). Our findings reveal no significant variance in the frequency or

memory B cell count between the optimized clonotypes and the remaining clonotypes.

Fig. 4 | Characteristics of 19 optimized clonotypes. a, Box plots showing the differences between optimized clonotypes and the remaining clonotypes for frequency (a), the number of memory B cells (b) and somatic hypermutation rate (c).

- As described in Figure 1c, the authors selected antibody sequences that had over 2% SHM. However, they do not discuss whether these antibodies were de novo elicited by the breakthrough infection or if they were already present from the prior vaccination and therefore recalled. This is an interesting point that should be addressed, ideally experimentally.

Response: Thank you for bringing up this question. Previous research has extensively discussed the immune response following vaccination and breakthrough infection. We have observed that memory B cells encoding IGHV2-5 and IGHV3-53/66 germlines exist in the vaccinated population at a relatively low percentage, while breakthrough infections would significantly elicit the expression of broad-spectrum neutralizing antibodies, as previously

described. (citation: Yuan et al. Structural basis of a shared antibody response to SARS-CoV-2. doi:10.1126/science.abd2321; Evans et al. Neutralizing antibody responses elicited by SARS-CoV-2 mRNA vaccination wane over time and are boosted by breakthrough infection. doi:10.1126/scitranslmed.abn8057; Walls et al. SARS-CoV-2 breakthrough infections elicit potent, broad, and durable neutralizing antibody responses. doi:10.1016/j.cell.2022.01.011.)

Additionally, since the relaxation of COVID-19 policies in December 2022, it has been estimated that large waves of mainland Chinese residents were infected with Omicron sub-variants, mostly BA.5.2 and BF.7. As a result, it has become exceedingly difficult for us to recruit another group of uninfected vaccine volunteers to experimentally test the above hypothesis. Furthermore, since our primary focus is on the discovery and characterization of antibodies following breakthrough infections, we opted not to overly emphasize this issue in this revision, with the reviewer's permission.

- On line 413, the authors write "...some elite antibodies present strong allelic preference" with a reference to Vanshylla et al. (ref 35). This does not appear to be the correct reference for this statement. Please explain.

Response: We apologize for this carelessness, this has been corrected in our revision.

- An additional issue with referencing relates to statements on line 98 and line 365, where the authors discuss immune imprinting. Reference 18 should be removed in both places as this is not a primary paper but a commentary.

Response: We apologize for this carelessness, this has been corrected in our

revision.

Reviewer #2 (Remarks to the Author):

In their manuscript, Guo et al investigate the antibody response in patients that had received multiple doses of the inactivated SARS-CoV-2 vaccine (BBBIP-CorV - Sinopharm, commonly used in China) after BA.1 breakthrough. They employ single cell sorting in combination with biophysical and EM/crystallography testing to identify broadly neutralising antibodies, and further develop them into antibody cocktails which they test in the K18-hACE2 model.

This is an interesting and well executed study which is suitable for publication in Nature Communications.

I have two minor comments:

1) Please expand the discussion, and in particular discuss how the use of the inactivated Sinovac vaccine may have affected the outcome of the study in comparison to the globally more widely used mRNA vaccines.

Response: We would like to express our gratitude to the reviewer for their positive evaluation of our work. While we have addressed the suggestions provided and expanded some of the discussion points as recommended, we have also taken into consideration that the comparison between different vaccines has been extensively discussed in previous research (Wang et al. doi:10.1038/s41586-022-04466-x; Cao et al. doi: 10.1038/s41422-021-00596-5; Vogel et al. doi:10.1038/s41586-021-03275-y.). As our focus is primarily on

the discovery and characterization of antibodies following breakthrough infection, we have opted not to overly emphasize this issue in our revision, with the reviewer's permission.

2) Please discuss, based on the reported mutations in XXB.1 and XXB.15, if the antibodies and cocktails here might also be applicable to these current VOCs. This could be addressed through structural modelling, or alternatively through RBD binding or viral assays.

Response: We thank this reviewer for raising this critical issue. In response, we have provided additional data in this revision on the neutralizing activities of TH027, TH132, TH272, and TH281 against emerging VOCs, such as BQ.1.1 and XBB. The neutralizing profile analysis revealed that all four antibodies maintained good neutralizing effects on BF.7 at single digit picomolar NT₅₀ values. Although BQ.1.1 and XBB have shown escape from TH027 and TH272, we are pleased to report that TH132, TH281, and the antibody cocktail TH027+TH132 exhibited broad-spectrum and potent neutralizing activity, with NT₅₀ values ranging from 13-36 ng/mL.

We have updated Figures 5, S10, and S11 in our revision accordingly.

Minor technical comment:

Line 421 "binding with the RBD" should presumably be "binding of the RBM (to ACE2)".

Response: We apologize for this carelessness, this has been corrected in our revision.

Reviewer #3 (Remarks to the Author):

The continued antigenic evolution of SARS-CoV-2 has led to numerous Omicron sublineages, which continue to drive waves of infection. These variants have escaped neutralization by all clinical-stage monoclonal antibodies (mAbs) due to spike protein receptor-binding domain mutations. Until recently, two FDA-approved mAb therapies (Bebtelovimab and Evusheld) were still available to treat individuals at high risk of progressing to severe COVID-19. However, recent data suggest that these are no longer effective against the most recent variants, such as XBB and BQ.1.1 (1). As such, there is a continued need to develop therapeutic monoclonal antibodies to combat SARS-CoV-2 infections. The manuscript by Guo et al describes several mAbs, obtained from BA.1 breakthrough-infected patients, capable of neutralizing D614G, Delta and several Omicron sublineages such as BA.2.12.1 and BA.4/5. This is a well-written, comprehensive study, but given the routine antibody discovery pipeline and highly competitive/fast-paced field of anti-SARS-CoV-2 antibodies, this paper does not offer much in terms of originality. Moreover, the manuscript is outdated with respect to the current pandemic situation. Conceivably, the mAbs described here could help to address the urgent clinical need for therapeutics to prevent or treat COVID-19. Still, their efficacy against more relevant Omicron variants needs to be demonstrated.

Main points to address

1. The ability to neutralize BA.2.12.1 and BA.5 is of diminishing relevance to the current pandemic situation. BQ.1.1, BA.4.6.3, XBB, and CH.1.1 can all escape neutralization by bebtelovimab, which also worked well against BA.4/5. Experimental evidence for efficacy against these VOCs should be performed

by live or pseudovirus neutralization assays. If any of these molecules can differentiate themselves from the long list of ineffective mAbs, it would demonstrate the relevance of the author's work and potential clinical application.

Response: We would like to express our appreciation to the reviewer for their positive evaluation of our work. In response to their feedback, we have included additional data in this revision on the neutralizing activities of TH027, TH132, TH272, and TH281 against emerging VOCs, specifically variants encoding the R346T mutation such as BF.7, BQ1.1, and XBB. The neutralizing profile analysis revealed that all four antibodies maintained good neutralizing effects on BF.7 at single digit picomolar NT₅₀ values. While TH027 and TH272 showed escape from BQ.1.1 and XBB. TH132, TH281, and the TH027+TH132 cocktail maintained a broad-spectrum and potent neutralizing activity against these VOCs.

It is worth noting that most of the neutralization evaluation data for the antibodies reported in this manuscript were completed before the emergence of new variants such as BF.7, BQ.1.1, XBB, and CH.1.1. Nevertheless, we are delighted to observe that the selected antibody pairs, using the strategies described in this manuscript, have maintained their potent neutralization capacity against the emerging VOCs.

In light of these findings, we have updated Figures 5, S10, and S11 in our revision accordingly. Thank you again for your valuable feedback.

2. Related to point one, if any of these mAbs can still neutralize currently circulating VOCs, the paper will require a rewrite to better place the results in the context of the current pandemic situation.

Reference

1) Cao et al, 2022: (<https://www.nature.com/articles/s41586-022-05644-7>)

Response: Thank you for your suggestion. In response, we have revised the corresponding paragraph accordingly.

REVIEWERS' COMMENTS

Reviewer #1 (Remarks to the Author):

The authors have addressed my concerns and the paper is consequently improved and more up to date.

Reviewer #2 (Remarks to the Author):

The authors have addressed all my questions.

Reviewer #3 (Remarks to the Author):

The authors have addressed my concerns and the manuscript is greatly improved as a result. The addition of new VOC neutralisation data demonstrates the potential value of TH027 + TH132 cocktail, which I hope can contribute to the prevention of SARS-CoV-2 infection in vulnerable patients.